# A non-canonical function for Centromere-associated protein-E controls centrosome integrity and orientation of cell division

Mikito Owa [1✉] & Brian Dynlacht [1✉]

Centromere-associated protein-E (CENP-E) is a kinesin motor localizing at kinetochores. Although its mitotic functions have been well studied, it has been challenging to investigate direct consequences of CENP-E removal using conventional methods because CENP-E depletion resulted in mitotic arrest. In this study, we harnessed an auxin-inducible degron system to achieve acute degradation of CENP-E. We revealed a kinetochore-independent role for CENP-E that removes pericentriolar material 1 (PCM1) from centrosomes in late S/early $G_2$ phase. After acute loss of CENP-E, centrosomal Polo-like kinase 1 (Plk1) localization is abrogated through accumulation of PCM1, resulting in aberrant phosphorylation and destabilization of centrosomes, which triggers shortened astral microtubules and oblique cell divisions. Furthermore, we also observed centrosome and cell division defects in cells from a microcephaly patient with mutations in *CENPE*. Orientation of cell division is deregulated in some microcephalic patients, and our unanticipated findings provide additional insights into how microcephaly can result from centrosomal defects.

[1] Department of Pathology, New York University Cancer Institute, New York University School of Medicine, New York, NY, USA. ✉email: Mikito.Owa@nyumc.org; Brian.Dynlacht@nyumc.org

The centrosome is an organizing center for microtubules (MTs) in metazoan cells. This organelle functions as a hub for MT-based protein transport during interphase, and it organizes spindle poles in mitosis. Within the centrosome, the two centrioles are embedded in pericentriolar material (PCM), which consists of γ-tubulin ring complexes, a nucleator of MTs, together with a cohort of proteins that regulate centrosome integrity[1]. Centrosomes and the environment surrounding this organelle are dynamically remodeled throughout the cell cycle. In interphase, centriolar satellites, dense granules that consist of multiple proteins essential for controlling centriole duplication, are scattered around centrosomes[2,3]. These granules are anchored on microtubules (MTs) through a scaffold protein, pericentriolar material 1 (PCM1; despite its name, this is not a component of the PCM). Beyond its ability to organize centriolar satellites on the MTs, PCM1 also prevents its interacting partners from being inappropriately relocated to centrosomes, as suggested by the observation that satellite proteins are constitutively localized at centrosomes after PCM1 depletion[4–8]. Centriolar satellites accumulate around centrosomes through dynein-driven transport during and after centriole duplication[9–11]. Subsequently, they are re-distributed within the cytoplasm as prophase commences[4,5]. On the other hand, Plk1 accumulates on the centrosome in prophase, where it phosphorylates PCM components, including pericentrin (PCNT) and Wdr62[12,13]. These modifications accelerate PCM expansion and formation of robust astral MTs, which anchor the cell cortex to maintain spindle orientation. Despite our knowledge of these dynamic events during prophase, the mechanisms underlying the dispersal of centriolar satellites—and its functional implications—remain obscure, and how PCM expansion is spatio-temporally regulated likewise remains unclear.

Recent proteomic studies have identified motor proteins that potentially interact with centriolar satellites[14,15], and centromere-associated protein-E (CENP-E) is one such candidate. Several studies previously reported that CENP-E is a kinesin motor localizing to kinetochores, where the protein reinforces MT-kinetochore interactions[16–20], and contributes to the spindle assembly checkpoint[21–23]. However, it is notable that most chromosomes were aligned at the metaphase plate even after CENP-E depletion[23]. In addition, CENP-E depletion in mammals led to mitotic arrest[21,22], and deletion of *CENPE* in mice resulted in early embryonic lethality[24], suggesting that the spindle assembly check point was satisfied without CENP-E in these species. Furthermore, all reported functions for CENP-E pertain to mitosis, during which time centriolar satellites are dispersed in the cytoplasm. Therefore, the functional consequences for inter-actions between CENP-E and centriolar satellite proteins, if any, remained elusive. In this study, we show that CENP-E has a non-canonical role around centrosomes in interphase. CENP-E removes PCM1 from the peri-centrosomal region in $G_2$ phase, and this transport is critical for structural stability of centrosomes and maintenance of spindle orientation in mitosis. Moreover, our findings can explain phenotypes associated with microcephaly, a developmentally related brain disorder, as evidenced by using patient-derived cells mutated in *CENPE*[25]. Thus, we have unveiled an unanticipated role for CENP-E in centrosome dynamics and have linked it to mechanisms that result in microcephaly.

## Results

**CENP-E is recruited around centrosomes in $G_2$ phase**. It was previously reported that CENP-E protein is initially synthesized prior to mitosis[16]. Western blotting of lysates from synchronized wild-type RPE-1 cells confirmed that cytoplasmic CENP-E levels were elevated in late $S/G_2$ phase, and they peaked in mitosis (Fig. 1a). However, in late $S/G_2$ phase, MTs are not attached to kinetochores, and therefore, whether cytoplasmic CENP-E had a

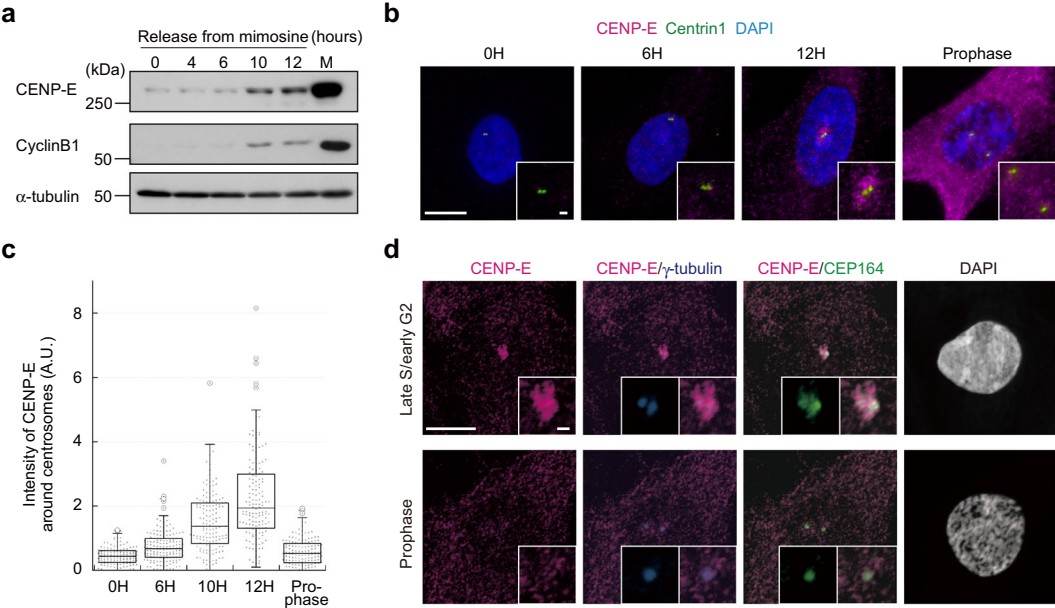

**Fig. 1 Cytoplasmic CENP-E is recruited around centrosomes in $G_2$ phase. a** Mimosine synchronized RPE-1 cells were released into fresh media and harvested 0, 4, 6, 10, and 12 h after mimosine removal. M phase cells were synchronized with monastrol. Total cell lysates were immuno-blotted with indicated antibodies. **b** RPE-1 cells released from mimosine were immuno-stained with antibodies against CENP-E (magenta) and centrin1 (green). Representative pictures 0, 6, and 12 h after release or in prophase are shown (scale bar = 10 μm). Inset in each panel is a magnified image around centrosomes (scale bar = 1 μm). **c** Signal intensities of CENP-E around centrosomes in **b** were measured and plotted ($N$ = 148, two independent experiments; whisker: 95% confidence interval; box: interquartile; center line: median). **d** Representative images for z-stacks around centrosomes in late S/early $G_2$ (top) or prophase (bottom) wild-type cells co-stained for indicated markers (scale bar = 10 μm). Inset in each panel is a magnified image around centrosomes (scale bar = 1 μm).

function in interphase was unclear. To understand its interphase role, we first explored CENP-E localization from interphase to mitotic onset using immunofluorescence. In early S phase, CENP-E signal was not detectable in the cytoplasm, in accordance with its low expression levels (Fig. 1b, c, 0 and 6 h). On the other hand, CENP-E was specifically enriched around centrosomes from late S to $G_2$ phase (Fig. 1b, c, 10 and 12 h; Fig. 1d, late S/early G2; Supplementary Fig. 1), and was dispersed throughout the cytoplasm by prophase (Fig. 1b–d, prophase). This pericentrosomal localization of CENP-E is consistent with recently identified interactions between centriolar satellites, typified by the PCM1 protein, and CENP-E[15]. Accordingly, PCM1 co-localized with CENP-E around centrosomes in late S/early G2 phase but not in prophase (Supplementary Fig. 2). Given that centriolar satellites are not localized at kinetochores or spindle poles during mitosis[4,5] (Supplementary Fig. 2), these data suggest that CENP-E has a previously uncharacterized role around the centrosome in interphase.

**Loss of CENP-E leads to cell cycle exit after one cell division cycle.** CENP-E depletion using siRNA or antibody injection led to

mitotic arrest[17,20,22]. However, these methods require extended periods of time for complete depletion or inactivation and depend on the specificity of siRNAs or antibodies. Therefore, it has been challenging to study the immediate consequences of CENP-E removal. To overcome these obstacles and investigate the pericentrosomal function of CENP-E, we established a conditional knockout (KO) cell line using gene-editing to biallelically introduce Auxin-inducible degrons at the endogenous locus (CENP-E-AID)[26–29]. One hour after auxin (IAA) addition, CENP-E was degraded and undetectable by western blotting (Fig. 2a; Supplementary Fig. 3b). Further, CENP-E signal was not detectable around centrosomes in interphase or at kinetochores in mitosis upon auxin treatment (Fig. 2b). Most mitotic CENP-E KO cells had misaligned chromosomes near spindle poles, a phenotype consistent with other CENP-E depletion studies[17,20,22,23,30] (Fig. 2b arrowheads; Supplementary Fig. 4a). In agreement with a previous report in mice[23], the complete loss of CENP-E did not lead to mitotic arrest but instead prompted a delay in pseudo-metaphase (Fig. 2c; Supplementary Fig. 3c). Importantly, nearly all CENP-E KO cells stopped growing and exited from the cell cycle after one cell division cycle (Fig. 2d, e). These CENP-E KO cells did not undergo apoptosis within this timeframe

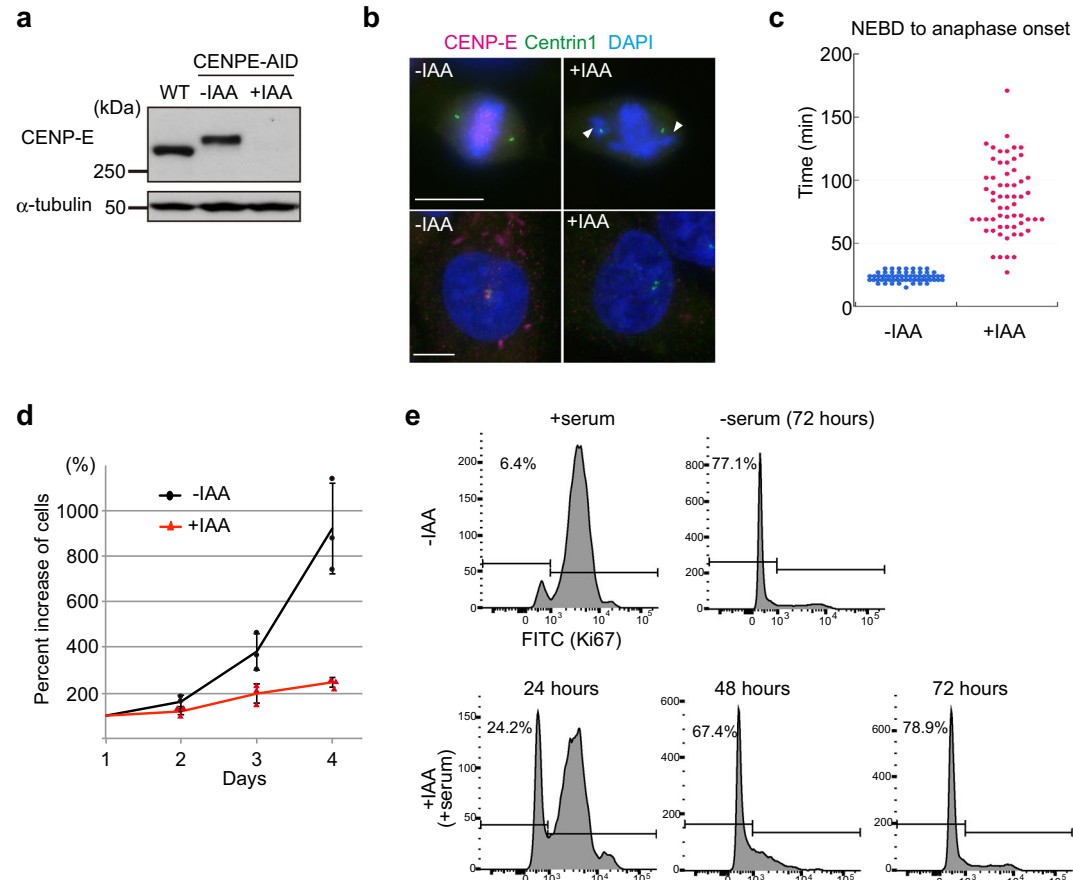

**Fig. 2 Loss of CENP-E induces cell cycle exit. a** Cell lysates from wild-type RPE-1 and CENP-E-AID (−IAA: not treated; +IAA: treated with IAA for 1 h) were immuno-blotted with indicated antibodies. The CENP-E band was shifted up in CENP-E-AID due to addition of AID and 3×FLAG tags. **b** Representative images of CENP-E-AID cells immuno-stained with antibodies against CENP-E (magenta) and centrin1 (green). Kinetochore-specific (top panels) and pericentriolar localization (bottom) of CENP-E diminished after IAA treatment. Arrowheads indicate misaligned chromosomes. Scale bar = 10 μm. **c** Time from NEBD to anaphase onset in CENP-E-AID cells with or without IAA was measured by live cell imaging and plotted (*N* = 60, three independent experiments). **d** Cell growth assays in CENP-E-AID cells with or without IAA. Each experiment commenced with ~10,000 cells (day1). Percent increases of cells were plotted (three independent experiments). **e** Proportions of cells that exited from the cell cycle were analyzed by FACS with Ki67-FITC staining. Asynchronous or serum-starved CENP-E-AID cells without IAA were used for positive or negative controls, respectively. Time on the histograms indicates duration of IAA treatment.

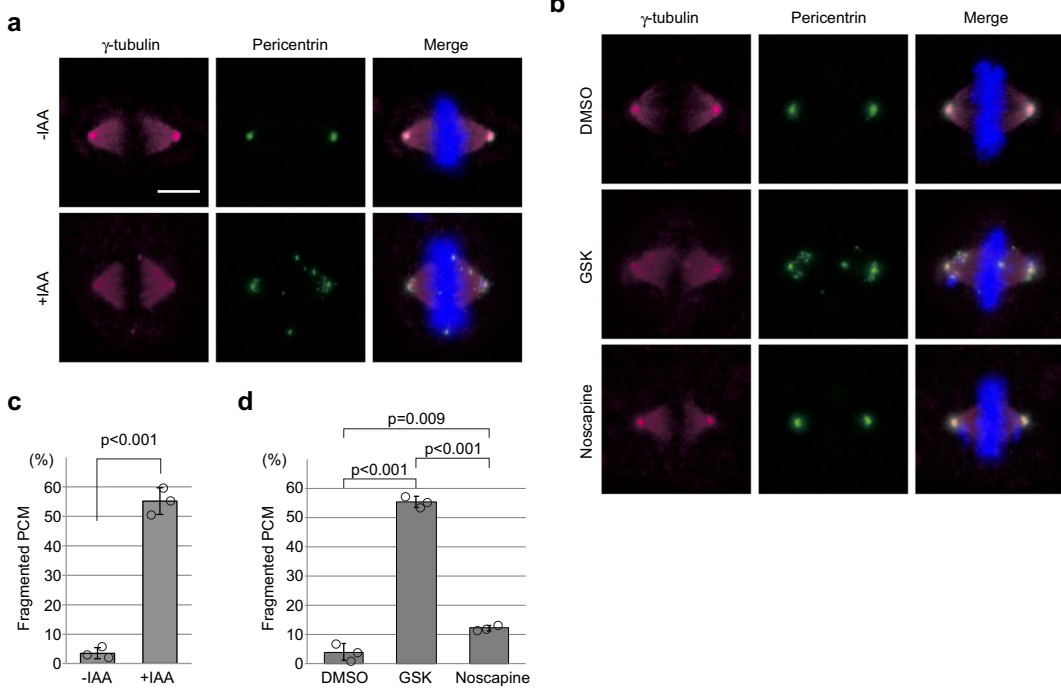

**Fig. 3 Loss of CENP-E or CENP-E inhibition leads to PCM fragmentation. a**, **b** Synchronized CENP-E-AID (**a**) or wild-type RPE-1 (**b**) cells with a single thymidine block were released into fresh media for 8 h with or without indicated drugs. The cells were then fixed and co-immunostained with antibodies against γ-tubulin (magenta) and PCNT (green). Representative images of metaphase or pseudo-metaphase cells in each sample are shown (scale bar = 10 μm). Percentages of cells with fragmented PCM are compared in the bar graph (**c**, **d**; >100 cells in total from three independent experiments; error bars: SD). *p*-values were calculated by an unpaired *t*-test (**c**) or Scheffe tests after ANOVA (**d**).

(Supplementary Fig. 3d). Loss of CENP-E led to chromosome mis-segregation in mice[23], suggesting that our CENP-E KO cells might exit from the cell cycle due to aneuploidy in daughter cells.

**Loss of CENP-E promotes PCM fragmentation and oblique cell divisions**. A previous study reported that spindle MTs were not properly focused at spindle poles in a small fraction of cells treated with CENP-E siRNA, despite exclusive localization of CENP-E at kinetochores in wild-type metaphase cells[22]. Focusing of spindle MTs at poles is dependent on dynein-driven transport along astral MTs, nucleated at the PCM[31]. Therefore, the possibility remained that unfocused spindle MTs in CENP-E-depleted cells arose from defects in PCM structure. Interestingly, by immuno-staining for PCM markers, we showed that loss of CENP-E also provoked severe defects in PCM morphology. In controls without auxin treatment, γ-tubulin and PCNT were sharply focused at each spindle pole (Fig. 3a, −IAA). On the other hand, γ-tubulin and PCNT foci were fragmented in 55% of CENP-E KO cells upon auxin addition (Fig. 3a, c, +IAA). Furthermore, cells treated with GSK923295 (GSK)[32], an inhibitor of CENP-E motor activity, also showed PCM fragmentation (55%; Fig. 3b, d, GSK), as well as misaligned chromosomes (Supplementary Fig. 4b), suggesting that fragmentation in CENP-E KO cells results from loss of CENP-E-dependent protein transport. A recent study suggested that prolonged pro-metaphase induced by drug treatments led to precocious centriole disengagement and PCM fragmentation in a separase-dependent manner[33]. To examine whether PCM fragmentation in the CENP-E KO was a consequence of mitotic delay, we treated RPE-1 with noscapine, a drug that induces mitotic arrest with misaligned chromosomes similar to CENP-E KO cells[34]. Noscapine treatment induced chromosome misalignments, as expected, as well as a slight increase in PCM fragmentation, but the percentage (12%; Fig. 3b, d, noscapine) was significantly lower than in GSK-treated cells

(55%), indicating that PCM fragmentation in CENP-E KO is not simply a by-product of mitotic delay or arrest. In addition, we did not observe a significant increase in separated centrioles after GSK treatment (Supplementary Fig. 5a). Instead, centrioles were misaligned with respect to the equator in 16% of metaphase CENP-E KO cells (Supplementary Fig. 5b). These results suggest that PCM fragmentation in the CENP-E KO was not driven by enhanced separase activity but rather that PCM was detached from centrioles.

Importantly, we also observed shortened astral MTs in our CENP-E KO cells (Fig. 4a, b). Previous studies showed that knock-down of *ASPM* and *Wdr62*, two genes implicated in microcephaly that encode mitotic spindle and centrosome proteins, also induced shortening of astral MTs in human cell lines, resulting in spindle mis-orientation[13,35,36]. Therefore, we performed live-cell imaging of cells with and without auxin treatment. We found that a high proportion of cells underwent rotation of their metaphase plates (50%; Fig. 4c, d; Supplementary Movie 1) with an oblique cell division angle (41%; Fig. 4c, e). *CENPE* mutations have also been implicated in microcephaly[25], and our results imply that analogous mechanisms could link the disease phenotypes arising from mutations in these three genes.

**PCM fragmentation in CENP-E KO is linked to aberrant Plk1 activity**. Since centrosomal Plk1 phosphorylates PCM components in prophase, and this phosphorylation is essential for robust PCM expansion[1,12], it was possible that PCM fragmentation in CENP-E KO reflected an aberrant modification of its constituents. To test this possibility, we first explored centrosomal Plk1 levels in prophase CENP-E-AID cells. By immunofluorescence, we revealed that centrosomal Plk1 levels in CENP-E KO cells were significantly diminished as compared to controls (Fig. 5a, b). We next investigated the impact of this Plk1 reduction on PCNT, as it is a key substrate and an essential factor for

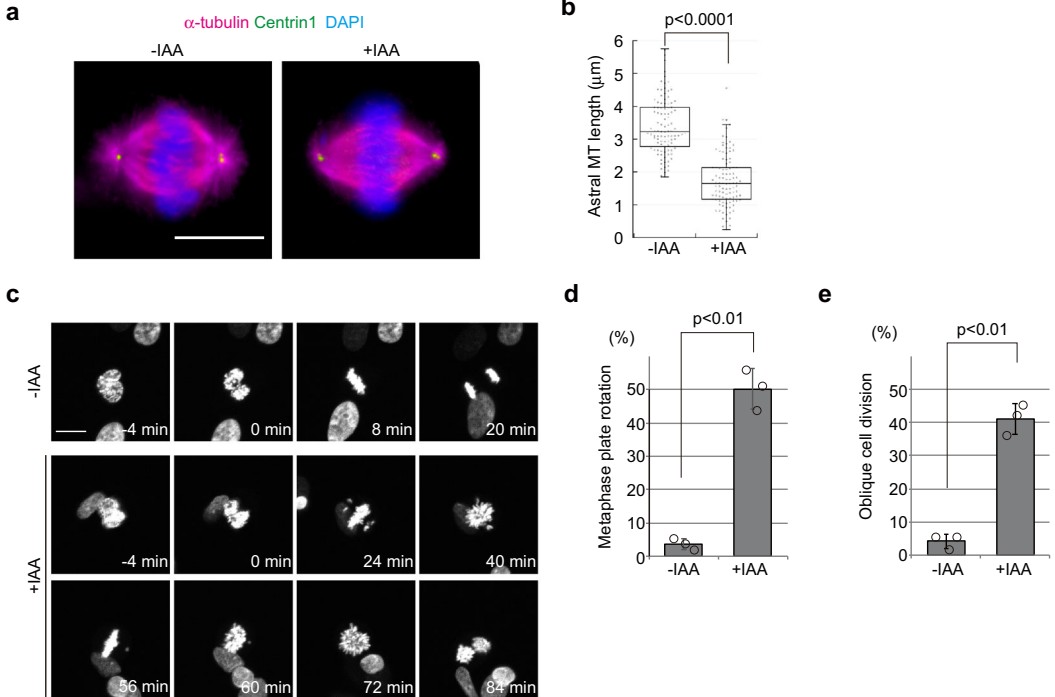

**Fig. 4 Loss of CENP-E induces shortened astral MTs and oblique cell divisions. a, b** Synchronized CENP-E-AID cells with a single thymidine block were released into fresh media for 8 h with or without indicated drugs and were co-immunostained with antibodies against α-tubulin (magenta) and centrin1 (green). Representative images of mitotic cells in each sample are shown (scale bar = 10 μm). Length of astral MTs was measured and plotted in **b** (N = 102, three independent experiments; whisker: 95% confidence interval; box: interquartile; center line: median). **c** Live-cell imaging of CENP-E-AID cells stably expressing H2B-GFP. Representative time-course images of control (−IAA) or CENP-E KO cells (+IAA) are shown (scale bar = 10 μm). **d** Percentages of cells in which the metaphase plate was rotated within 1 h after NEBD are shown in the bar graph (>100 cells in total from three independent experiments; error bars: SD). **e** Percentages of oblique cell divisions (see methods for definition) are shown in the bar graph (>100 cells in total from three independent experiments; error bars: SD). *p*-values were calculated by a Mann–Whitney *U* test (**b**) or unpaired *t*-tests (**d**, **e**).

PCM expansion[12]. Previous studies have identified sites on PCNT phosphorylated by Plk1, including S1241[12], which is essential for PCM expansion. After immuno-staining with a phospho-specific antibody (PCNT-pS1241)[12], we found that the fluorescence intensity of pS1241 was significantly lower on centrosomes in prophase CENP-E KO cells as compared to controls, similar to our findings with Plk1 (Fig. 5c, d). Importantly, however, total PCNT intensity on centrosomes was not reduced in prophase CENP-E KO cells (Supplementary Fig. 6), suggesting that the reduced pS1241 intensity was not due to defects in centrosomal PCNT accumulation but instead reflected the reduced Plk1 recruitment.

We further carried out PCNT cleavage assays to examine C-terminal phosphorylation of PCNT by Plk1[37]. Treatment with ZM447439 (ZM, an inhibitor of Aurora kinases) results in forced exit of mitotically arrested cells from M phase, and PCNT is cleaved in a Plk1-phosphorylation dependent manner. Although we did not detect the cleaved form of PCNT, most likely due to rapid degradation after cleavage, full-length PCNT was nearly undetectable in ZM-treated controls replete with CENP-E, suggesting near-complete cleavage during mitotic exit (Fig. 5e, −IAA). As a control, and as described previously[37], we showed that this cleavage was substantially blocked by inhibition of Plk1 through BI2536 (BI) treatment, attesting to its dependence on Plk1 (Fig. 5e, compare lanes 2 and 6). Importantly, similar to Plk1-inhibited cells, ZM-treated CENP-E KO cells retained full-length PCNT, at levels approximating those in untreated cells (Fig. 5e, +IAA; compare lanes 3 and 4). Thus, we conclude that Plk1 fails to efficiently phosphorylate PCNT in CENP-E KO cells.

Finally, to determine whether the reduction in Plk1 levels could explain the PCM defects, we ectopically expressed Plk1 in CENP-E KO cells. Notably, over-expression of constitutively active Plk1 (Plk1-T210D) in CENP-E KO rescued PCM fragmentation, whereas expression of the catalytically inactive protein (Plk1-K82R) did not (Fig. 5f, g). Moreover, Plk1-T210D expression also rescued reductions in PCNT-pS1241 phosphorylation and shortened astral MTs (Supplementary Fig. 7). In striking contrast, chromosome misalignment was not rescued by over-expression of Plk1-T210D (Supplementary Fig. 4c). These data strongly suggest that (1) the reduction of Plk1 levels on centrosomes in CENP-E KO cells results in PCM fragmentation and (2) PCM defects in CENP-E KO cells are independent of kinetochore dysfunction associated with chromosome misalignment.

**CENP-E prevents PCM1 accumulation around centrosomes prior to mitosis**. As described above, CENP-E is recruited around centrosomes in late S/early G2 phase, during which time centriolar satellites are re-distributed[4]. Since CENP-E is a processive kinesin that moves toward the plus end of MTs[17], we hypothesized that CENP-E transports PCM1 from the vicinity of centrosomes, as it is redistributed throughout the cytoplasm by the initiation of prophase. We therefore investigated the distribution of PCM1 around centrosomes in late prophase/early pro-metaphase cells. In control cells, PCM1 exhibited the expected centriolar satellite-like localization, with PCM1 foci scattered around the centrosome (Fig. 6a, −IAA). In contrast, PCM1 was tightly concentrated within the immediate vicinity of centrosomes in CENP-E KO cells (Fig. 6a, +IAA, c). Importantly,

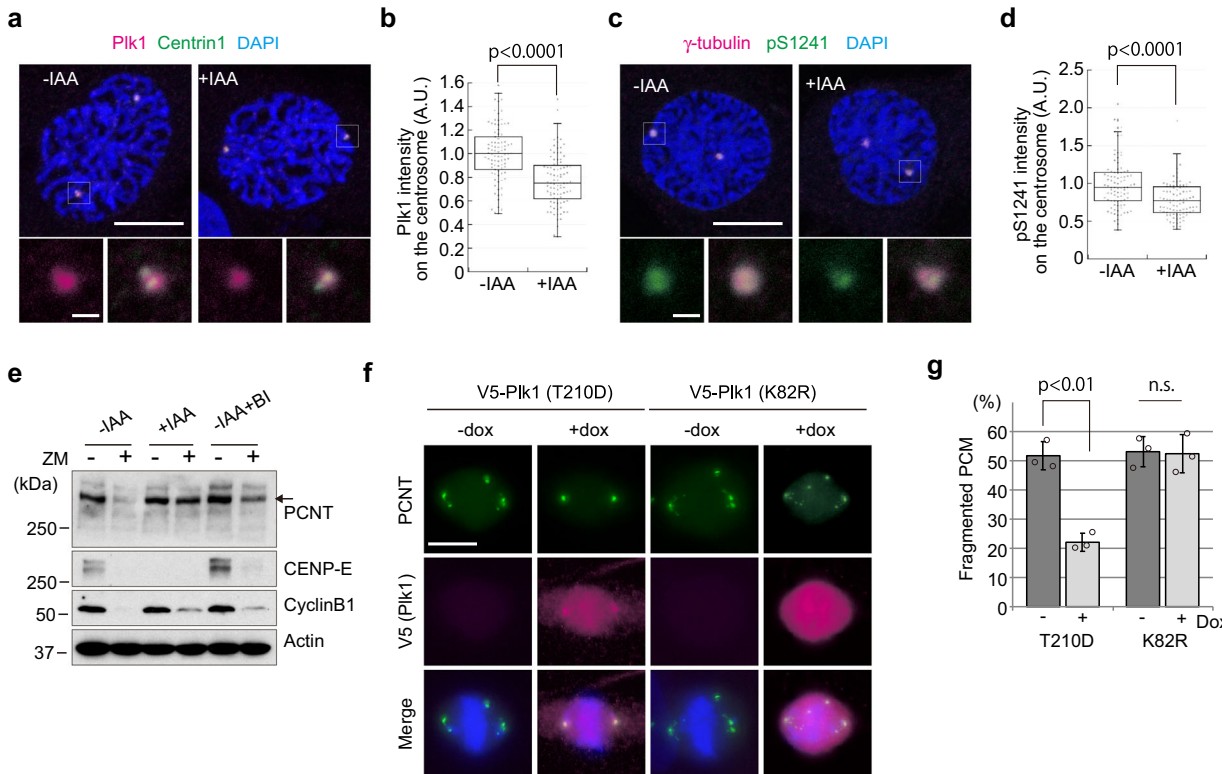

**Fig. 5 PCM1 fragmentation in CENP-E KO results from reductions in centrosomal Plk1 levels and aberrant phosphorylation of PCNT. a**, **b** Representative images for IAA-treated or un-treated CENP-E-AID cells in prophase co-immunostained with antibodies against Plk1 (magenta) and centrin1 (green; **a**, scale bar = 10 μm). The area enclosed by the square in each image is magnified and shown under the panel (scale bar = 1 μm). Relative Plk1 intensities on the centrosome in prophase cells were plotted in **b** ($N = 90$, three independent experiments; whisker: 95% confidence interval; box: interquartile; center line: median). **c**, **d** Representative images for IAA-treated or untreated CENP-E-AID cells in prophase co-immunostained with antibodies against γ-tubulin (magenta) and PCNT-pS1241 (green; **c**, scale bar = 10 μm). The area enclosed by the square in each image is magnified and shown under the panel (scale bar = 1 μm). Relative PCNT-pS1241 intensities on the centrosome in prophase cells were plotted in **d** ($N = 90$, three independent experiments; whisker: 95% confidence interval; box: interquartile; center line: median). **e** Control (−IAA), CENP-E KO (+IAA), and Plk1 inhibited (−IAA+BI) cells synchronized with 100 nM paclitaxel were shaken off from plates and forced to exit from mitosis with 2 μM ZM44739 for 1 h. Cell lysates were immuno-blotted with indicated antibodies. The arrow indicates full-length PCNT. **f**, **g** CENP-E-AID cell lines stably expressing doxycycline (dox)-inducible V5-tagged Plk1 (T210D: constitutively active; K82R: catalytically inactive) were synchronized with thymidine for 22 h, and released for 8 h. The cells were stained with antibodies against V5-tag (magenta) and PCNT (green). In +dox samples, 1 μg/ml dox was maintained throughout the experiments. Representative images for metaphase cells in each sample are shown (**f**, scale bar = 10 μm). Percentages of metaphase cells with fragmented PCM are compared in **g** (>100 cells in total from three independent experiments; error bars: SD). *p*-values were calculated by Mann–Whitney *U* tests (**b**, **d**) or an unpaired *t*-test (**g**).

the MT network was not affected by loss of CENP-E in this stage of the cell cycle (Supplementary Fig. 8), indicating that the PCM1 accumulation was not due to defects in microtubular tracks for transport, but rather was due to loss of CENP-E itself. We further tested whether CENP-E inhibition with GSK leads to PCM1 accumulation around centrosomes in late prophase/early pro-metaphase. Similar to CENP-E KO cells, GSK treatment induced centrosomal PCM1 accumulation (Fig. 6b, d, GSK). This effect was reversible, since subsequent removal of GSK allowed the redistribution of PCM1 within the cytoplasm (Fig. 6b, d, GSK washout). These data suggest that CENP-E removes PCM1 from the peri-centrosomal region by prophase, during which time Plk1 phosphorylates PCM proteins.

As described above, a recent proteome-wide study identified interactions between CENP-E and PCM1, although this interaction was not explored in detail[15]. To determine whether these interactions reflected the observed CENP-E-dependent PCM1 redistribution and to clarify the cell cycle stage in which CENP-E interacts with PCM1, we performed immunoprecipitations through the carboxy-terminal FLAG epitope of CENP-E-AID protein using lysates from cells synchronized by treatment with

mimosine (late $G_1$ phase) or monastrol (M phase). PCM1 co-precipitated with CENP-E in samples released from mimosine for 6 and 10 h (Fig. 6e, 6 h and 10 h), and the amount of co-precipitated PCM1 increased in accordance with CENP-E levels. On the other hand, PCM1 was not detected in precipitates from lysates of cells released for 12 h (Fig. 6e, 12 h) or from monastrol-synchronized cells (Fig. 6e, M). Given that most cells reach $G_2$ phase by 10 h after release from mimosine (Supplementary Fig. 1), these data demonstrate that CENP-E interacts with PCM1 most likely in late S/early $G_2$ phase, consistent with the idea that CENP-E transports PCM1 and centriolar satellites during this period.

**PCM1 depletion rescues PCM-related defects in CENP-E KO.** In CENP-E KO cells, peri-centrosomal PCM1 accumulation coincided with Plk1 reduction at centrosomes in prophase (Figs. 5a and 6a). Since PCM1 is thought to restrict centriolar satellite proteins from being recruited to centrosomes[4–8], it was conceivable that the accumulation of PCM1 affected Plk1 localization on centrosomes in KO cells. Therefore, we performed

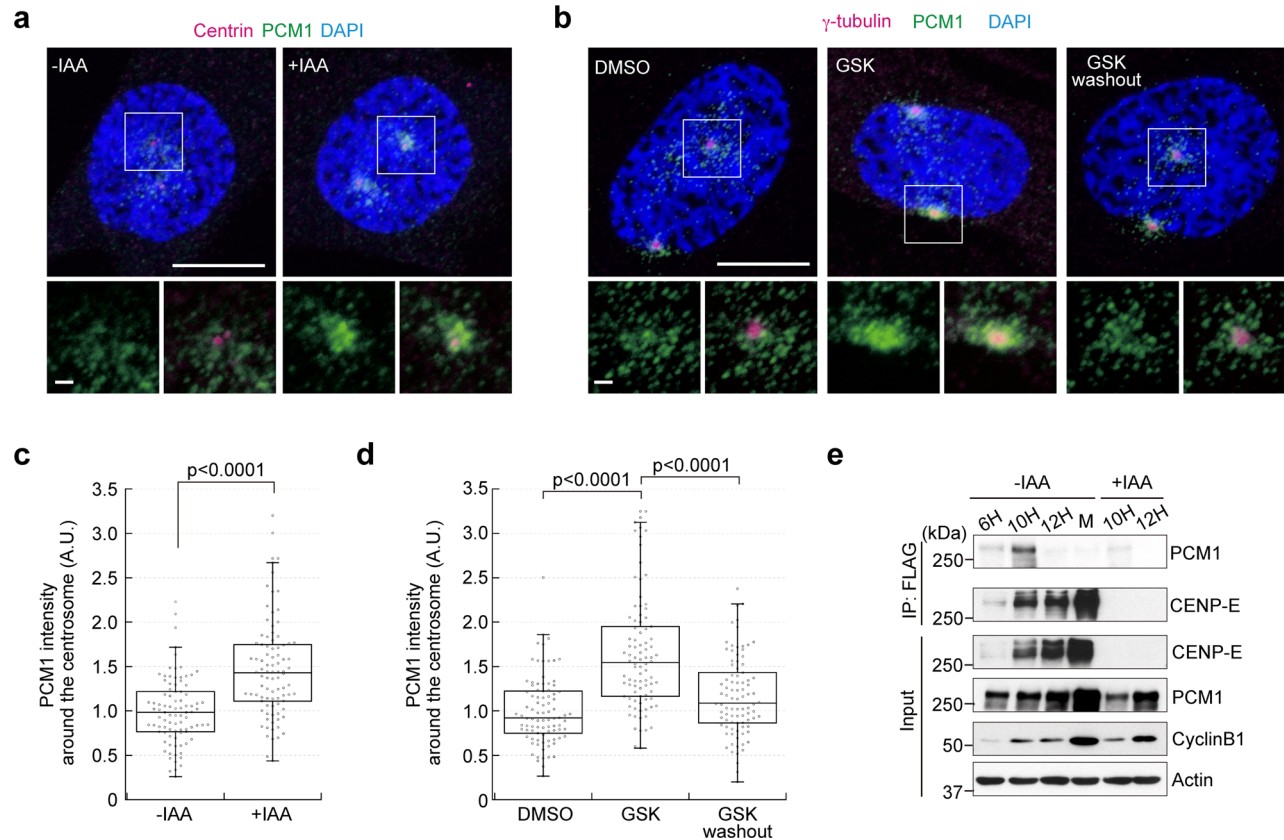

**Fig. 6 Loss of CENP-E induces accumulation of PCM1 around centrosomes in prophase. a, b** CENP-E-AID (**a**) or wild-type RPE-1 (**b**) cells synchronized with a single thymidine block were released into fresh media for 8 h with or without indicated drugs. The cells were then fixed and co-immunostained with antibodies against centrosome markers (magenta) and PCM1 (green). For GSK washout, the drug was removed and cells were released into fresh media in the last 1 h. Representative images for cells in late prophase/early pro-metaphase were shown (scale bar = 10 μm). The area enclosed by the square in each image is magnified and shown under the panel (scale bar = 1 μm). Relative PCM1 intensities around the centrosome were plotted (**c**, **d**; N = 90, three independent experiments; whisker: 95% confidence interval; box: interquartile; center line: median). **e** Cell lysates from mimosine (6 h, 10 h, and 12 h, indicating hours after release) or monastrol (M) synchronized CENP-E-AID cells were immunoprecipitated with FLAG-M2 beads. 10 h and 12 h (+IAA) samples were for negative controls. The precipitates were immuno-blotted with anti-CENP-E and anti-PCM1 antibodies (left). Inputs in each sample were also blotted with indicated antibodies (right). p-values were calculated by Mann–Whitney U tests.

PCM1 ablation experiments in CENP-E-AID cells. We found that PCM1 depletion rescued the reduction in centrosomal Plk1 provoked by loss of CENP-E (Fig. 7a, b), suggesting that the accumulation of PCM1 in CENP-E KO cells is directly responsible for perturbing Plk1 recruitment to centrosomes.

Importantly, and consistent with the recovery of centrosomal Plk1 levels, siPCM1 treatment reversed the reduction in PCNT phosphorylation at S1241 observed in CENP-E KO cells (Fig. 7c, d). Consequently, PCM1 depletion also rescued both PCM fragmentation (Fig. 7e) and shortening of astral MTs caused by the loss of CENP-E (Fig. 7f). These data strongly suggest that the PCM-related phenotypes in CENP-E KO cells stem from the accumulation of PCM1 around centrosomes triggered by the absence of a motor protein that normally moves it away from the organelle. Importantly, chromosome misalignment in the CENP-E KO was not rescued by PCM1 depletion (Supplementary Fig. 4d), indicating that the PCM1-Plk1 pathway is not involved in kinetochore defects caused by the loss of CENP-E.

**Patient LCLs mutated in *CENPE* exhibit PCM and cell division defects**. Heterozygous mutations in *CENPE* (D933N/K1355E) have been identified in two siblings with microcephalic primordial dwarfism[25]. These mutations are located within the coiled-coil region in the middle of the protein but do not affect

*CENPE* expression levels. Moreover, as the mutations did not result in embryonic lethality, lymphoblastoid cell lines (LCLs) derived from these patients have been successfully established, providing further evidence that disease mutations in *CENPE* do not lead to cell cycle exit, in contrast with our CENP-E KO cells. To examine whether the mutations drive PCM-related defects observed in the CENP-E KO, we explored PCM morphology by immunofluorescence using LCLs from one of the patients and an apparently healthy donor (control). Although PCM fragmentation was rarely observed in the patient LCLs, centrosomal PCNT levels in metaphase were significantly decreased compared to control cells (Fig. 8a, b), suggesting that the mutations in *CENPE* affect PCM expansion and/or maintenance of PCM structures. Consistently, centrosomal pS1241 levels also significantly decreased in late prophase/early pro-metaphase patient cells (Fig. 8c, d). In agreement with the morphological defects in patient PCM, live-cell imaging revealed that chromosomes were obliquely segregated into daughter cells in 41% of patient LCLs, whereas the percentage of oblique cell divisions in control cells was 13% (Fig. 8e, f). These data indicate that the heterozygous mutations in *CENPE* in microcephaly patients lead to PCM defects resulting in oblique cell divisions, which is a typical phenotype associated with the disease. Importantly, patient LCLs also exhibited PCM1 accumulation around centrosomes in late prophase/early pro-metaphase (Fig. 8g, h), reminiscent of

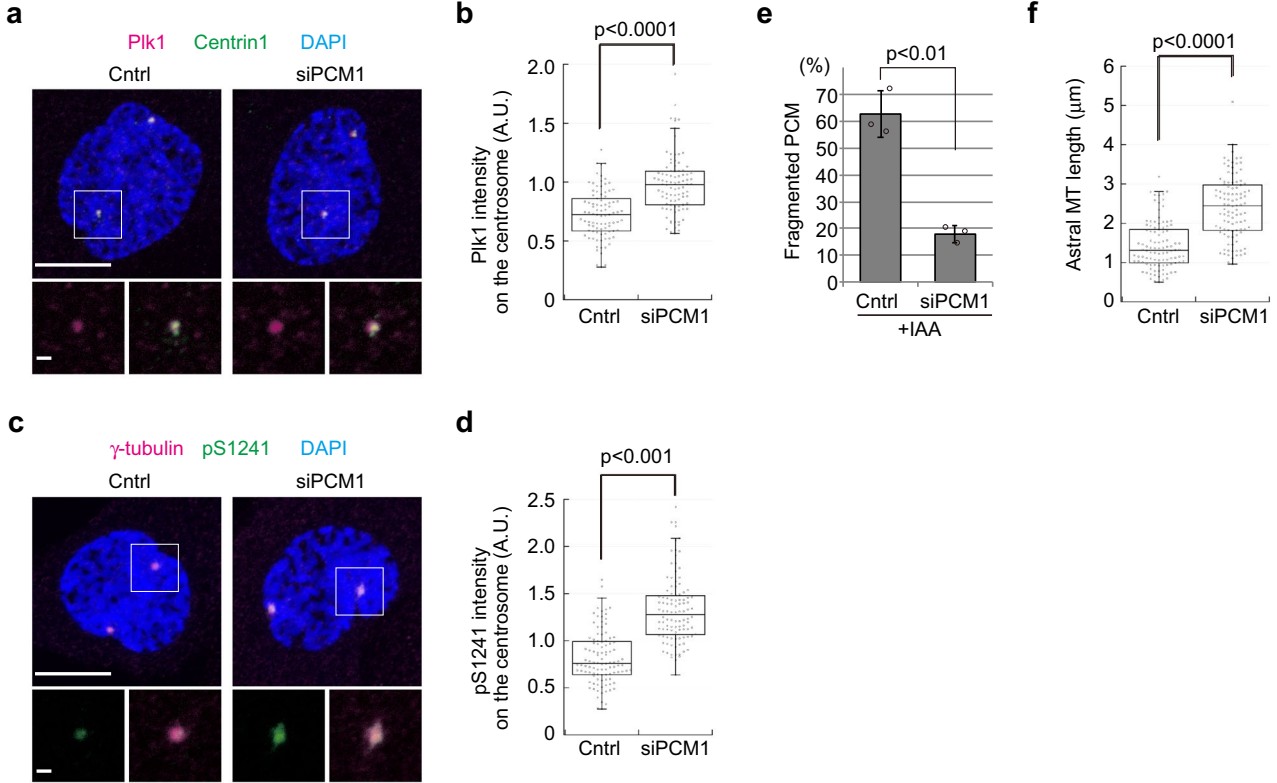

**Fig. 7 Accumulation of PCM1 around centrosomes perturbs centrosomal Plk1 recruitment and PCNT phosphorylation in CENP-E KO. a**, **b** CENP-E-AID cells were synchronized with thymidine for 22 h. In the first 8 h of the synchronization, cells were treated with control siRNA or siPCM1. The cells were then released for 8 h with IAA, fixed, and co-immunostained with antibodies against Plk1 (magenta) and centrin1 (green). Representative images for prophase cells in each sample are shown (**a**, scale bar = 10 μm). The area enclosed by the square in each image is magnified and shown under the panel (scale bar = 1 μm). Relative Plk1 intensities on centrosomes in prophase cells were plotted in **b** ($N = 90$, three independent experiments; whisker: 95% confidence interval; box: interquartile; center line: median). **c**, **d** Samples prepared as **a** were co-immunostained with antibodies against γ-tubulin (magenta) and PCNT-pS1241 (green). Representative images for prophase cells in each sample are shown (**c**, scale bar = 10 μm). The area enclosed by the square in each image is magnified and shown under the panel (scale bar = 1 μm). Relative PCNT-pS1241 intensities on the centrosome in prophase cells were plotted in **d** ($N = 104$, three independent experiments; whisker: 95% confidence interval; box: interquartile; center line: median). **e**, **f** CENP-E-AID cells were synchronized with siRNA treatments as **a**. The cells were then released for 8 h with IAA, and percentages of PCM fragmentation (**e**) or length of astral MTs (**f**) was analyzed and plotted as in Fig. 3a, c, respectively (>100 cells in total from three independent experiments; error bars: SD). *p*-values were calculated by Mann–Whitney *U* tests (**b**, **d**, **f**) or an unpaired *t*-test (**e**).

CENP-E KO or GSK-treated RPE-1 cells. Furthermore, PCM1 was co-precipitated with CENP-E from control lysates, whereas PCM1 protein was not detectable in immunoprecipitates from patient lysates (Supplementary Fig. 9). These data indicate that D933N/K1355E mutations affect CENP-E-dependent PCM1 redistribution, resulting in PCM and cell-division defects in patient LCLs.

## Discussion

CENP-E is a kinesin motor that transports chromosomes to the metaphase plate along spindle MTs. Persistent, complete loss of CENP-E causes delays in metaphase, most likely due to incomplete chromosome alignment, followed by cell cycle exit (Fig. 2c–e; Supplementary Fig. 3b)[22,23]. Therefore, it has been challenging to investigate direct consequences of CENP-E removal using siRNA or conventional KO cell lines. Here, acute degradation of CENP-E in our AID cell line enabled us to identify and characterize a second, distinct function for CENP-E apart from its role at kinetochores (Fig. 9). Namely, we found that in $G_2$ phase, CENP-E transports PCM1 from the vicinity of centrosomes to a location occupied by centriolar satellites. This redistribution of PCM1 is important to maintain the level of centrosomal Plk1 in prophase, when it phosphorylates PCNT to

facilitate PCM expansion[12,37]. Loss of CENP-E leads to reductions in phosphorylation levels of PCM, resulting in PCM fragmentation, loss of astral MTs, and oblique cell divisions. We propose that CENP-E represents the first kinesin identified as a transporter of centriolar satellites, moving them away from the centrosome. However, we note that 45% of CENP-E KO cells retained a normal PCM appearance (Fig. 3c), implying that another kinesin or MT remodeling in mitosis could also contribute to the redistribution of PCM1 cooperatively or redundantly with CENP-E.

A recent study reported that centromere dysfunction causes dispersion of PCM during mitosis[38]. Since CENP-E is a component of the kinetochore, it was possible that this effect may partially overlap phenotypes caused by the loss of CENP-E. However, the dispersion upon kinetochore dysfunction was limited to the vicinity of spindle poles, in contrast to PCM fragmentation caused by loss of CENP-E or its inhibition (Fig. 3a, b). Furthermore, and importantly, the PCM-related phenotypes observed in CENP-E KO cells were rescued by over-expression of active-Plk1 or PCM1 depletion (Fig. 5f, g, Supplementary Fig. 7, and Fig. 7), whereas chromosome misalignment, a phenotype induced by kinetochore dysfunction, was not rescued through these manipulations (Supplementary Fig. 4c, d). These findings indicate that the second, interphase function for CENP-E is

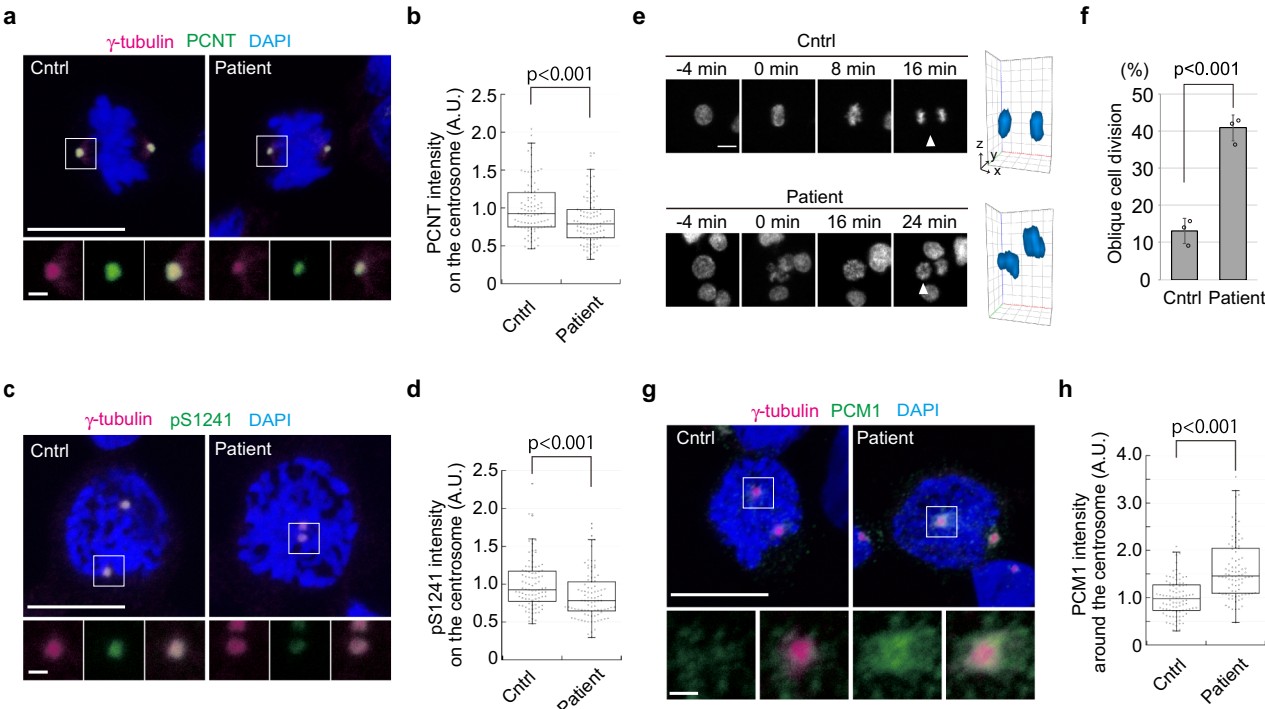

**Fig. 8 LCLs from a microcephalic patient with *CENP-E* mutations show PCM defects and oblique divisions. a, b** Control (Cntrl) and patient LCLs were synchronized with 2 mM thymidine for 22 h, and released for 5 h. Representative images for metaphase cells co-immunostained with antibodies against γ-tubulin (magenta) and PCNT (green) are shown (**a**, scale bar = 10 μm). The area enclosed by the square in each image is magnified and shown under the panel (scale bar = 1 μm). Relative PCNT intensities on the centrosome in metaphase cells were plotted in **b** (N = 90, three independent experiments; whisker: 95% confidence interval; box: interquartile; center line: median). **c, d** Representative images for prophase LCLs co-immunostained with antibodies against γ-tubulin (magenta) and PCNT-pS1241 (green; **c**, scale bar = 10 μm). The area enclosed by the square in each image is magnified and shown under the panel (scale bar = 1 μm). Relative PCNT-pS1241 intensities on the centrosome in prophase cells were plotted in **d** (N = 90, three independent experiments; whisker: 95% confidence interval; box: interquartile; center line: median). **e** Live-cell imaging of control (Cntrl) and patient LCLs. DNA was visualized with Hoechst 33342. Representative time-course images of control (top) or patient (bottom) LCLs are shown (scale bar = 10 μm). 3D reconstructions in the last time points (viewed from arrowheads) are also shown next to the images. **f** Percentages of oblique cell divisions observed in the live-cell imaging (see Methods section for definition) were compared in the bar graph (>100 cells in total from three independent experiments; error bars: SD). **g, h** Representative images for prophase LCLs co-immunostained with antibodies against γ-tubulin (magenta) and PCM1 (green; **g**, scale bar = 10 μm). The area enclosed by the square in each image is magnified and shown under the panel (scale bar = 1 μm). Relative PCM1 intensities around the centrosome in prophase cells were plotted in **h** (N = 90, three independent experiments; whisker: 95% confidence interval; box: interquartile; center line: median). p-values were calculated by Mann–Whitney U tests (**b**, **d**, **h**) or an unpaired t-test (**f**).

functionally separable from its role at kinetochores, and the interphase pathway we propose in our model is most critical for maintenance of PCM integrity during mitosis.

It was recently reported that depletion of autophagy-related proteins caused elevated levels and abnormal accumulation of centriolar satellite proteins, including PCM1[39]. The accumulation leads to formation of ectopic centrin or γ-tubulin foci, resulting in multipolar spindles. On the other hand, despite PCM fragmentation and assembly of abnormal astral MTs after CENP-E removal, nearly all KO cells formed bipolar spindles. In addition, we did not observe centrin amplification, indicating that PCM1 accumulation in CENP-E KO was near physiological levels as normally observed in late S/$G_2$ wild-type cells[4]. During bipolar spindle formation, centrosomes are subject to pulling and pushing forces generated by dynein and kinesin-5, respectively[40]. Loss of spindle pole integrity causes centrosome fragmentation in $G_2$/M transition, resulting in multipolar spindles. Nevertheless, our live-cell imaging revealed that all of the CENP-E KO cells formed a metaphase plate after NEBD (Fig. 4c; Supplementary Movie 1). Furthermore, the fragmented PCM in CENP-E KO cells did not capture chromosomes, suggesting that the PCM foci were not ectopically formed by accumulated PCM1 around centrosomes but, more likely, dissociated from centrosomes after bipolar spindle formation due to aberrant phosphorylation of its components. Indeed, reduction in PCNT phosphorylation after loss of CENP-E was moderate (Fig. 5c), and the observed phosphorylation levels could thus be sufficient for centrosome maturation. It is possible that the mild phenotype enables centrosomes in CENP-E KO cells to endure the mechanical stress and pass through the $G_2$/M transition without PCM fragmentation. Spindle MTs are cross-bridged to each other by a large number of MT-associated proteins, such as a NuMA-dynein-dynactin complex[41], kinesin-14[41], TPX2[42,43] and a clathrin-mediated complex[44,45], and these MT-crosslinks stabilize the structure at bipolar spindles. However, antagonistic motors still exert opposite forces between spindle poles and chromosomes after bipolar spindle formation[46,47], and those may induce PCM fragmentation in CENP-E KO cells.

We observed shortening of astral MTs in CENP-E KO cells (Fig. 4a, b), which may be a consequence of PCM fragmentation after bipolar spindle formation. Since ASPM or WDR62 depletion induces spindle mis-orientation together with astral MT defects[13,35], oblique cell divisions in CENPE-KO (Fig. 4c–e and Supplementary Movie 1) could likewise be due to abnormal astral

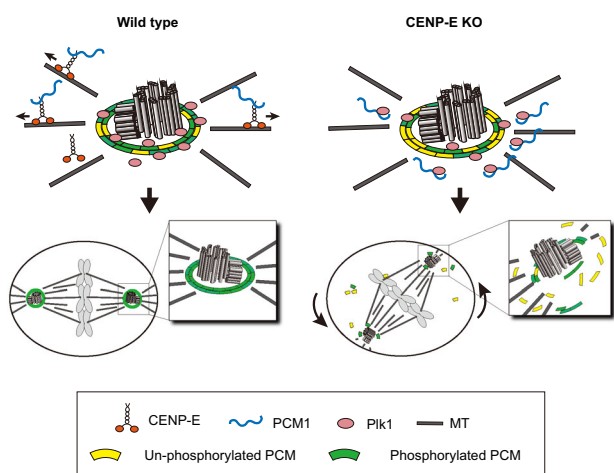

**Fig. 9 A model summarizing results in this study.** Interphase CENP-E removes PCM1 from the vicinity of the centrosome by prophase, during which time Plk1 phosphorylates PCM. In the CENP-E KO, PCM1 remains around the centrosome even in prophase, Plk1 is sequestered by PCM1, and phosphorylation levels of PCM decline. Insufficiently phosphorylated PCM is fragmented during mitosis, resulting in chromosome mis-orientation and oblique divisions.

MTs. On the other hand, a previous study reported that mis-aligned chromosomes lead to cortical displacement of LGN in a kinetochore-localized Plk1-dependent manner[48]. Since LGN recruits NuMA to the cortex[49], deregulation of cortical LGN results in spindle positioning defects. Since chromosome mis-alignment is a major phenotype in CENP-E KO, it is possible that cortical LGN localization is altered by loss of CENP-E. However, since patient LCLs also exhibited oblique cell divisions, it is likely that spindle mis-orientation in CENP-E KO can be attributed to shortening of astral MTs.

To date, 25 genes have been implicated in autosomal recessive primary microcephaly (MCPH; OMIM phenotypic series: PS251200)[36,50], and *CENPE* (MCPH13) is one of these genes. Although *PCNT* is not included among the MCPH genes, mutations in the gene cause microcephalic osteodysplastic primordial dwarfism type II (MOPDII)[51–53], suggesting strong correlations between PCM defects and microcephaly. Since CENP-E was first identified as a component of the kinetochore involved in chromosome alignment, it was anticipated that the mutations in *CENPE* provoke microcephaly through defects in chromosome segregation. However, CENP-E patient cells do not exhibit cell cycle exit seen in our CENP-E KO, suggesting that CENP-E mutant proteins in patients retain partial activity essential for cell cycle progression. These data therefore suggest a new mechanism for diseases associated with CENP-E mutations. In patient LCLs, we observed phenotypes reminiscent of our CENP-E KO cells, such as aberrant phosphorylation of PCNT and oblique cell divisions. Instead of PCM fragmentation, however, we observed reductions in the mass of PCM in patient LCLs (Fig. 8a, b). This may reflect cell-type specific differences wherein fragmented PCM was quickly dispersed in the cytoplasm in LCLs. Another possibility is that the phosphorylation level of PCNT was insufficient for proper centrosome maturation in patient LCLs, but centrosomes had undergone sufficient (albeit partial) maturation to enable bipolar spindle formation. Nevertheless, our data suggest that centrosomal defects in patient LCLs drive oblique cell divisions, as observed in the CENP-E KO. As described above, ASPM (MCPH5) and WDR62 (MCPH2) also enforce

proper spindle orientation through astral MT organization[13,35]. Although it is necessary to investigate whether and how oblique cell divisions trigger cell cycle exit or premature differentiation during brain development, our findings suggest a generalized mechanism for control of cell division orientation, which is deregulated during brain development in some microcephalic patients.

Thus, our findings have clarified an unexpected and non-canonical function for interphase CENP-E—separable from its role at kinetochores—that is critical for maintenance of PCM structure, astral MTs, and spindle orientation, akin to the function of other genes mutated in microcephaly.

## Methods

**Cell culture**. hTERT-RPE1 cells were cultured in Dulbecco's Modified Eagle's Medium: Nutrient Mixture F12 (DMEM/F12; Corning, 10–092-CV) supplemented with 10% fetal bovine serum (FBS) and 50 U/ml penicillin/streptomycin (P/S; Corning, 30–001-CI). Control and patient LCLs were cultured in Roswell Park Memorial Institute Medium (RPMI 1640; Corning, 10–040-CV) supplemented with 15% FBS and P/S (50U/ml). RPE-1 cells were obtained from the American Type Culture Collection (ATCC). Control (LR09–293m) and patient (LR05–054a2) LCLs were gifts from Dr. G. Mirzaa (Seattle Children's Hospital) and were obtained with patient's consent. All cells were cultured at 37 °C with 5% CO$_2$ and routinely tested with the Universal Mycoplasma Detection Kit (ATCC, 30–1012 K).

**Production of CENP-E-AID cell line with CRISPR/Cas9**. For the CENP-E-AID cell line, an sgRNA (5′-TTGGCACGCCTCCTCAGGCA-3′) was sub-cloned into PX458 (Addgene, 48138). The C-terminal region of the *CENP-E* gene was amplified with the following primers; 5′-TGCCCAAACTGGCCTTGAAC-3′, 5′-CCAAAGAGCCGAGAATGACTTGA-3′. The amplicon was integrated into pBluescript SK(−) digested with EcoRV. A template vector which carries full-length AID-3×FLAG-P2A-BSD was produced with the backbone of pMK392 (Addgene, 121193). Full-length AID-3×FLAG-P2A-BSD was integrated into the site just before the terminal codon of the sub-cloned *CENP-E* gene. In total, 250 ng of each of the donor and Cas9-sgRNA vectors were transfected to RPE-1 cells cultured in six wells with 1.5 μl of FuGENE® HD (Promega, E2311) overnight. Transfected cells were then selected with blasticidin S (5 μg/ml) for 10 days, and biallelic mutations in the clones were validated by PCR, followed by DNA sequencing. ARF16-PB1-HA-P2A-OsTIR1 from pMGS46 (Addgene, 126580) was transferred to pCDH-CMV-EF1a-Neo. Retrovirus was produced with the vector and infected to the CENP-E-AID clones, followed by selection with G418 (500 μg/ml).

**Cell cycle synchronization**. To increase the number of mitotic cells, RPE1 cells and LCLs were treated with 3 mM or 2 mM thymidine (Sigma, T9250) for 22 h, respectively. The RPE1 cells were released for 8 h and the LCLs for 5 h before fixation. For synchronization at late G1 phase, RPE-1 cells were treated with 0.4 mM mimosine (Sigma, M0253) for 22 h. Mimosine was washed out twice with fresh media. For synchronization in mitosis, RPE-1 cells were first treated with 3 mM thymidine for 22 h. After washing out thymidine, the cells were incubated with 5 μM paclitaxel (Selleck, S1150) for 12 h or 100 μM monastrol for 9 h. For inhibition of Plk1 activity, cells synchronized with paclitaxel were treated with 100 nM BI2536 (Selleck, S1109) for 3 h. For forced cell cycle exit, mitotic cells were treated with 2 μM ZM447439 (Selleck, S1103) for 1 h. In all, 0.5 mM IAA (Sigma, I5148) for CENP-E KO cells, 50 μM noscapine (Sigma, 363960), or 200 nM GSK923295 (APExBIO, A3450) for wild-type cells was added right after release from mimosine or thymidine.

**Immunofluorescence microscopy**. RPE-1 cells or LCLs on coverslips were fixed with cold methanol at −20 °C for 5 min, washed twice with PBS, and blocked with 3% donkey serum in 0.1% Triton X-100/PBS for 15 min. The coverslips were then incubated with primary antibodies for 120 min at room temperature. Cells were washed with 0.1% Triton X-100/PBS three times, and the coverslips were incubated with secondary antibodies for 90 min at room temperature, nuclei were stained with DAPI, and coverslips were mounted with ProLong™ Diamond Antifade Mountant (Invitrogen, P36961). Cells were observed using an Axiovert 200 M (×63, NA 1.4, Zeiss) or an LSM800 (×63, NA 1.4, Zeiss). Maximum intensities in Z-stack images were projected to a single image for figures. Signal intensities were measured after subtracting background intensities using ImageJ. For LCLs, coverslips were coated with 0.1% polyethylenimine (PEI), followed by fibronectin (50 μg/ml; Sigma, F2006). Antibodies used for IF are listed in Supplementary Table 1.

**Live-cell imaging**. H2B-GFP was introduced into CENP-E-AID cells using retroviral delivery as described above, followed by selection with 5 μg/ml of puromycin. Cells expressing H2B-GFP were seeded onto glass bottom dishes (Greiner,

627870) and treated with 3 mM thymidine for 22 h. In total, 7 h after thymidine wash-out, cells were observed using an LSM800 confocal microscope (×20, NA 0.8, Zeiss) operated by Zen software (Zeiss). LCLs were synchronized with 2 mM thymidine for 22 h. In all, 1 h after release from thymidine, cells were seeded onto glass bottom dishes coated with PEI, and incubated with 2 μM Hoechst33342 for 15 min. After changing media, cells were observed using the same equipment and conditions described above. Z-stack (thickness of a slice was 4 μm) images were acquired every 4 min, and multiple slices were projected to one image for figures using ImageJ. We defined cells as obliquely dividing if the center of segregated chromosomes were in different Z-stacks in the first frame of anaphase.

**Immunoprecipitation and western blotting**. Cells were harvested and resuspended in lysis buffer (30 mM Tris-HCl pH 7.5, 150 mM NaCl, 0.5% NP-40, 2 mM EDTA, 2 mM EGTA, 1 mM DTT, 10 mM NaF, 50 mM β-glycerophosphate, 5% glycerol and protease inhibitors) on ice for 15 min, followed by centrifugation at $16,000 \times g$ for 15 min. For immunoprecipitation, supernatants were mixed with ANTI-FLAG® M2 Affinity Gel (Sigma, A2220) and rotated at 4 °C for 3 h. The beads were washed with lysis buffer four times, and proteins were eluted with 0.1 M Glycine-HCl pH 2.8. In SDS-PAGE, 50 μg of lysates were loaded into each lane, and proteins were transferred to nitrocellulose membranes (GE Healthcare, 10600007). Antibodies used for western blots are listed in Supplementary Table 1, and uncropped blots are shown in Supplementary Fig. 10.

**FACS analysis and cell proliferation assays**. For FACS analysis, harvested cells were fixed with 70% ethanol for 5 min at −20 °C, washed twice with PBS containing 1% BSA, and then stained with Propidium iodide (PI). All data were acquired with an LSRII UV cell analyzer (BD Bioscience) and analyzed by FlowJo software. For cell proliferation assays, 10,000 cells were seeded in 3.5 cm dishes and harvested at indicated time-points. Cells were counted with a Vi-Cell Cell Viability Analyzer (Beckman Coulter).

**Statistics and reproducibility**. Statistical tests, $n$ values, and the number of times that the measurements were repeated were described in figure legends.

**Reporting summary**. Further information on research design is available in the Nature Research Reporting Summary linked to this article.

## Data availability
Data supporting the findings of this manuscript are available from the corresponding authors upon reasonable request. Source data for all graphs are available. For LCLs, please contact Dr. G. Mirzaa (Seattle Children's Institute).

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

## Acknowledgements

We are grateful to Drs. M. Berkseth and G. Mirzaa (Seattle Children's Institute) for providing control and *CENPE* mutant patient LCLs and K. Rhee (Seoul National University) for providing antibodies against PCNT. We thank Masahide Kikkawa (The University of Tokyo), Masafumi Hirono (Hosei University), and Ken-ichi Wakabayashi (Tokyo Institute of Technology) for critical reading of the manuscript. B.D. was supported by NIH grant 5R01GM120776–09. M.O. was supported by a JSPS Overseas Research Fellowship.

## Author contributions

M.O. and B.D. designed experiments, and M.O. performed all the experiments. M.O. and B.D. wrote the manuscript.

## Competing interests

The authors declare no competing interests.
