## [Peer Review File · Communications Biology]

Reviewers' comments:

Reviewer #1 (Remarks to the Author):

CENP-E is known as a kinesin motor localizing to kinetochores, where the protein reinforces MT-kinetochore interactions and contributes to the spindle assembly checkpoint. In this manuscript, the authors investigated roles of CENP-E outside kinetochores using the CENP-E-AID construct. First, they observed that CENP-E was specifically enriched around centrosomes and centriolar satellites in interphase. In CENP-E KO cells, spindle poles were fragmented, astral MTs were shortened, and PCM-1 was concentrated near the centrosomes. Since CENP-E transports PCM1 from the vicinity of centrosomes to cell periphery, CENP-E KO results in centrosomal accumulation of Plk1. As a result, upregulated centrosomal Plk1 triggers loss of astral microtubules and oblique cell divisions. Finally, the authors observed centrosome and cell division defects in cells from a microcephaly patient with mutations in CENP-E, providing evidence that microcephaly can result from centrosomal defects in the CENP-E mutants. They propose that CENP-E represents the first kinesin identified as a transporter of centriolar satellites, moving them away from the centrosome.

It is significant that the manuscript reports novel functions of CENP-E outside kinetochores. The authors claim that the phenotypes are raised from non-kinetochore CENP-E. Nevertheless, it still remains to be cleared whether the reported phenotypes are solely attributed by the nonkinetochore CENP-E or not.

Follows are major points.

- Fig. 1b: They claim that CENP-E physically interacts with PCM1. It is worth to coimmunostain interphase and mitotic cells with the CENP-E and PCM1 antibodies.
- Fig. 4e: Immunoblot analysis of PCNT reveals two bands at least; an intact and a cleaved band. An intact band is abundant at G2/M phase while a cleaved band becomes predominant after mitotic exit. The immunoblot data in Fig. 4e are not as expected.
- Fig. 5c: The authors conclude that CENP-E physically interacts with PCM1, based on coimmunoprecipitation of endogenous PCM1 with ectopic FLAG-CENP-E. Their conclusion has to be supported by additional experimental evidence, such as definition of the interacting domains and GST pull-down assays. In addition, it is odd that FLAGCENP-E did not coimmunoprecipitate PCM1 in the 12-H cells and monastrol-treated cells.
- Fig. 7a: They may examine fragmentation of the spindle poles in the patient cells.

Reviewer #2 (Remarks to the Author):

Owa and Dynlacht set out to study the kinetochore-independent function of CENP-E, a kinesin motor with a well documented role at the kinetochore during mitosis. Previous proteomic studies suggest that CENP-E may interact with centriolar satellite proteins and thus could function outside of kinetochores. However, studying kinetochore-independent function of CENP-E has been challenging because prolonged loss of CENP-E activity leads to mitotic arrest. Using an auxin-inducible degron system, the authors were able to acutely deplete CENP-E few hours before mitosis and observed several cellular phenotypes during mitosis (e.g., PCM fragmentation, abnormal astral microtubules, spindle misorientation, cell cycle exit). They concluded that the major contributor of these phenotypes

is the aberrant centrosomal accumulation of centriolar satellite protein PCM1 during mitosis. They propose that without CENP-E transporting PCM1 away from the centrosome before mitosis, PCM1 untimely accumulated at mitotic centrosomes, leading to insufficient centrosomal PLK1 levels and the subsequent hypophosphorylation of PCNT, a PCM protein critical for establishing mitotic PCM. Because similar PCM and cell division defects were observed in the lymphoblastoid cell lines of a microcephalic patient with compound heterozygous CENPE mutations, they propose that these common PCM-related defects contribute to microcephaly.

This manuscript presents a set of experiments that are of commendable quality to address the kinetochore-independent function of CENP-E. The demonstration of CENP-E in regulating PCM1 localization and affecting PCM integrity is novel. However, whether CENP-E is indeed the motor that transports PCM1 is not sufficiently supported by the data. The significance of observed cellular defects in contributing to microcephaly is also not definitely established. Overall, this is a well constructed study, but the strength of some claims needs to be toned down or the claims need to be further supported by additional data. The following comments should help the authors revise this manuscript.

Major concerns

1. To exclude that the IAA treatment alone may contribute to the observed mitotic phenotypes, control experiments such as doing the same IAA treatment in the cells expressing OstTIR1 with wild-type untagged CENP-E or in the cells with another unrelated degron-tagged protein should be performed (e.g., doing a cell cycle profiling as done in Extended Data Fig. 1).
2. Although compelling genetic evidence has linked centrosome dysfunction to primary autosomal recessive microcephaly, the molecular mechanism underlying the etiology of this disease remains elusive. Studies over the years suggest that premature differentiation and subsequent depletion of neural stem cells (e.g., due to spindle misorientation), apoptosis, cell cycle arrest, neural precursor delamination, or some combinations of these, could be the underlying causes. However, increasing evidence suggests that spindle orientation defects alone cannot explain microcephaly (e.g., Insolera et al., 2014, Jayaraman et al., 2016). Therefore, claiming misorientation of cell division/mitotic spindles upon CENP-E loss as a “unifying principle that explains how microcephaly can result from centrosome defects” is not well justified.
3. The authors observed that in CENP-E knockout cells, PCM1 failed to disperse into the cytoplasm in prophase. They thus conclude that CENP-E transports PCM1 away from the centrosome prior to mitosis. Although this is a very reasonable speculation and CENP-E and PCM1 can be in the same complex (the co-IP experiments), there is no direct evidence to support that CENP-E is indeed the motor that transports PCM1. Other indirect effect of CENP-E on PCM1 localization should also be considered. Or additional data should be obtained to show the direct transport of PCM1 by CENP-E along microtubules.

Minor points

1. Fig. 1b, c: How did the authors determine the time point 12H as G2 phase? Is CENP-E really dispersed from the centrosome in prophase? Or does the increase of cytoplasmic CENP-E in prophase “mask” the CENP-E signals around the centrosome as CENP-E appears to be distributed widely in the cytoplasm, including the pericentrosomal regions?
2. Fig. 2: Can the misaligned chromosomes be the result of compromised CENP-E function at BOTH the centrosome and kinetochore? Although the authors emphasize the “interphase role” of CENP-E, most of the described phenotypes were observed in M phase, where the kinetochore function of CENP-E is expected to be compromised as well.

3. What happened after cell cycle arrest upon CENP-E loss? Does this arrest eventually trigger apoptosis? Apoptosis of neural progenitors has been linked to the etiology of microcephaly. It thus would be interesting to know if apoptosis follows cell cycle arrest upon CENP-E loss.

4. Lines 118-121: CENP-E loss could lead to several different mitotic phenotypes, e.g., PCM fragmentation with unfocused spindle poles (Fig. 3a), abnormal astral MTs but with focused spindle poles (Fig. 3c). The time-lapse experiment did not include a MT marker. So it is not clear whether the cells undergoing rotation of metaphase plates are the cells with abnormal astral MTs. This statement should be modified.

5. The reduction of PLK1 level and S1241 phosphorylated PCNT at the centrosome is very modest. The authors should comment on why the very modest changes could lead to profound mitotic defects such as fragmented PCM.

6. Fig. 4f: The panels of K82R and T210D conditions should be switched so they will be in the same order as described in the text and in Fig. 4g. In addition, K82R/+dox samples are not the best representative images because the PCNT fragmentation phenotypes are not obvious and seemed to be partially rescued.

7. Does PLK1(T210D) overexpression also rescue PCNT S1241 phosphorylation, PCNT cleavage defects, astral MT defects, and oblique cell division caused by CENP-E depletion?

8. Fig 5. The authors should remind readers that CENP-E-AID fusion contains FLAG tags.

9. Lines 212 and 228: "Compound heterozygous mutations" should be a more proper term in describing the D933N/K1355E mutations.

10. The central piece of the proposed model in this study revolves around the aberrant centrosomal PCM1 accumulation during mitosis upon compromised CENP-E activity, which in turn leads to PCM and cell division defects. The authors should determine if the defects in centrosomal PCNT level and PCNT S1241 phosphorylation in the patent LCLs are indeed caused by the aberrant accumulation of PCM1 at the centrosome. If yes, can PCM1 knockdown also rescue the phenotypes as does in CENP-E KO cells?

11. Lines 239-246: As mentioned previously, this conclusion is not supported by the data; there is no direct evidence to show that CENP-E transports PCM1 away from the centrosome during G2 phase.

12. All characterized CENP-E KO phenotypes are in mitosis. The data did not determine whether these phenotypes are indeed caused by compromised CENP-E function during interphase. Claiming that this kinetochore-independent function of CENP-E is exerted in interphase as an ¹¹SEP¹¹ interphase pathway" is not well justified.

13. The size of the scale bar in Extended Data Fig. 3b is not indicated.

Reviewer #3 (Remarks to the Author):

In the manuscript entitled "A non-canonical function for Centromere-associated protein-E (CENP-E) controls centrosome integrity and orientation of cell division", Owa and Dynlacht established an auxin-inducible degron system to acutely degrade CENP-E at a specific cell cycle stage in order to study its

potential non-kinetochore functions. The authors show that CENP-E localizes at the centrosome in G2 phase and that its removal leads to accumulation of PCM1 in prophase, associated with decreased centrosomal levels of Plk1 and consequent reduction in pericentrin phosphorylation. They conclude that CENP-E normally transports PCM1 away from the centrosomes and thus prevents the negative effects of PCM1 accumulation, which otherwise sequesters Plk1 from the centrosomes and diminishes pericentrin phosphorylation. They propose that this reduction in pericentrin phosphorylation impairs astral microtubules and promotes centrosome fragmentation in mitosis, finally leading to problems in spindle positioning.

Although identification of a non-kinetochore function of CENP-E would be of importance and interest, the results presented here are still quite preliminary and therefore are not sufficient to support the main conclusions. There are several issues that would need to be addressed prior to the manuscript publication.

Major points

1) related to Figure 3 – The authors suggest that PCM fragmentation is a consequence of the inability of CENP-E to remove PCM1 away from the centrosome. The data presented here do not exclude the possibility that PCM fragments due to mitotic arrest. To strengthen the conclusions made upon the data presented in Ext. Data 3a, the authors should have measured the intercentriolar distance. However, the best way to address whether this effect was a consequence of mitotic arrest would be to perform live-cell imaging using fluorescently labeled gamma-tubulin or pericentrin.

The effect on astral microtubules should be quantified in a less subjective manner, e.g. by measuring the fluorescence intensity levels. The observation that spindles rotate in the absence of CENP-E (and other proteins important for chromosome congression) is well studied and explained by the effect that the uncongressed chromosomes make on cortical LGN-NuMA-Dynein axis via kinetochore-localized Plk1 (Tame et al., EMBO Rep 2016).

2) related to Figure 4 – It is important to clarify the cause of reduced levels of centrosomal Plk1. The data presented here do not exclude a possibility that the reduced Plk1 levels are a consequence of problems with centrosome maturation. To properly address this, the fluorescence intensities of gamma tubulin and pericentrin should be quantified in these cells. In addition, microtubule nucleation assays should be performed, such as microtubule regrowth assay and quantification of EB1/3 dynamics in interphase and mitosis, which would additionally clarify the impact of CENP-E removal on astral microtubules.

3) related to Figure 5 – The authors conclude that CENP-E removes PCM1 away from the centrosomes and that the lack of CENP-E-dependent transport promotes PCM1 accumulation in the vicinity of centrosomes. To dissect whether the observed PCM1 accumulation depended on CENP-E mediated transport, as proposed in the study, or it depended purely on centrosomal localization of CENP-E, an experiment with CENP-E inhibition should be performed. CENP-E inhibition affects the activity of CENP-E, while it should not affect its localization. Thus, if this treatment provides a similar PCM1 accumulation, one could conclude that this is transport-dependent.

The authors show that CENP-E interacts with PCM1 in G2, but not in mitosis. However, all the results are shown in mitotic cells (prophase and metaphase). Do the authors observe any centrosome-related problems in G2?

4) related to Figure 6 – The authors conclude that the accumulated PCM1 observed in CENP-E-depleted cells sequesters Plk1 from centrosomes. If true, Plk1 should colocalize with PCM1, which does

not seem to be the case.

As mentioned above, the effect on astral microtubules should be quantified in a less subjective way (fluorescence intensity, EB1/3 dynamics).

5) related to Figure 7 – The link between the patient-derived CENP-E mutant cell line phenotype and the model proposed in this study is poorly established. In contrast to the data obtained by CENP-E degradation, these cells do not show centrosome fragmentation. A slight decrease in pericentrin levels is shown in prophase, but it is not clear whether PCM1 levels and localization is affected, which otherwise constitutes the main phenotype in this study. There is no data provided about this particular CENP-E mutant. The lack of chromosome congression problems suggest that its kinetochore function is intact, but does it localize to centrosome in the same way as the wild type? Does it interact with PCM1?

Minor points

There are several omissions related to the citation of relevant literature throughout the manuscript. Here are a few examples:

- 1) PCM1 accumulation was previously shown to have a critical impact on centrosome stability, leading to mitotic centrosome fragmentation and unbalanced chromosome segregation (Grønbaek Holdgaard et al., Nat Comms 2019).
- 2) As mentioned above, the spindle rotation phenotype in presence of uncongressed chromosomes is explained via its Plk1-dependent effect on cortical LGN-NuMA-Dynein axis (Tame et al. EMBO Rep 2016).
- 3) The possibility of PCM fragmentation by microtubule-induced forces (the discussion section) should be discussed in the context of earlier studies (reviewed in Maiato and Logarinho, NCB 2014).

Reviewer #4 (Remarks to the Author):

The manuscript by Owa and Dynlacht investigates a non-canonical, kinetochore-independent, interphase function of the kinesin motor CENP-E. Through the development of an acute auxin-mediated degradation of CENP-E they investigate the role of CENP-E in the dispersal of centriolar satellites typically observed at the G2/M transition, building on previous data in which CENP-E was found to interact with the prototype centriolar satellite marker PCM1. First, they show that CENP-E localizes "around" the centrosomes in G2 and was dispersed in mitotic prophase. They report that acute CENP-E degradation (aprox. 1h) caused the expected mitotic phenotypes previously reported after CENP-E loss of function, and note that, in mitotic cells, PCM was fragmented in about 50% of the knockdown cells, together with loss of astral microtubules and spindle rotation. The observed PCM fragmentation after acute CENP-E degradation was accompanied by a decrease in Plk1 and one of its phosphorylated substrates at centrosomes. Overexpression of active Plk1 rescued PCM fragmentation after CENP-E knockdown. In prophase, PCM1 was apparently enriched at the centrosomes after CENP-E knockdown. Immunoprecipitation experiments revealed that CENP-E likely interacts with PCM1 in G2, but not in mitosis. Co-depletion of CENP-E and PCM1, rescued PCM-related defects in prophase cells. Lastly, the authors used a lymphoblastoid cell line from a human donor with microcephalic primordial dwarfism with an heterozygous CENP-E mutation and show that PCM (PCNT) was also reduced in mitotic cells. However, they failed to observe PCM fragmentation in the patient material. Live-cell imaging confirmed a significant increase in cells with oblique divisions, as compared to material from a healthy donor. Altogether, the authors propose a model in which a non-canonical interphase role of CENP-E regulates PCM structure and may account for defects commonly observed in microcephaly patients. Overall, I find this study potentially interesting, as it explores a poorly

understood aspect of CENP-E function in interphase (G2). However, I have a substantial number of concerns that must be addressed in order to support publication of the present work.

Major issues:

1- In the proposed model, CENP-E microtubule plus-end-directed motor activity would be required to disperse centriolar satellites as cells transit from G2 to mitotic prophase. This conclusion is entirely based on a single experiment (fig 5a,b) that suggests higher accumulation of PCM1 at centrosomes after acute CENP-E-depletion. However, the example shown is clearly not a prophase cell (I would confidently classify it as an early prometaphase cell). I understand that the authors have quantified this in a number of cells, but they must ensure that equivalent and comparable images were used in these quantifications, ideally with proper markers that unequivocally distinguish both states (e.g. including a nuclear envelope marker to confirm that the cell is indeed in prophase). The proposed model also implies that CENP-E must somehow overcome the activity of the microtubule-minus-end directed motor Dynein, which is required to concentrate centriolar satellites near the centrosome (see Kubo et al., JCB, 1999). Importantly, Dynein and CENP-E activities are differentially regulated by α -tubulin detyrosination: it inhibits the initiation of Dynein/Dynactin processivity (see e.g. McKenney et al., EMBO J. 2016), while promoting CENP-E motor activity (see Barisic et al., Science, 2015). This means that in order for CENP-E to eventually overcome Dynein, microtubules used in the dispersion of centriolar satellites in prophase must be enriched in detyrosinated α -tubulin. Is this the case? Please show and discuss.

2- Given that essential microtubule nucleation factors, such as gamma-tubulin, disperse after acute CENP-E depletion, it is expected that microtubule nucleation capacity at centrosomes will be affected (as shown for astral MTs in mitosis). As so, was it the absence of astral microtubules (e.g. due to defective centrosome maturation) that prevented the normal dispersal of centriolar satellites by any CENP-E-independent means, or was it specifically the lack of CENP-E motor activity? This could be assessed by allowing centrosomes to mature in G2 and develop normal astral microtubules and then acutely inhibit CENP-E motor activity with the GSK inhibitor in prophase cells.

3- It is arguable that the observed mitotic phenotype results from a direct role of CENP-E in centriolar satellite dispersal, leading to PCM expansion/maturation. Indeed, loss of centrosome integrity and pole splitting, as well as spindle rotation were not rescued by co-depletion of PCM1 and CENP-E, suggesting that these downstream mitotic effects are unrelated to a possible role of CENP-E in PCM organization. Moreover, after acute CENP-E depletion, most cells remain in mitosis for at least 1h, up to 2h. A mitotic delay imposed by unrelated cellular perturbations (including CENP-E inhibition with GSK) is known to generate cohesion fatigue, often followed by loss of centrosome integrity and pole splitting, as well as spindle rotation (see Daum et al., Curr Biol, 2011; Stevens et al., PLoS ONE, 2011). Thus, the authors must exclude that the observed loss of centrosome integrity with consequent pole splitting and spindle rotation is not simply a consequence of the observed mitotic delay imposed by acute CENP-E depletion, rather than through PCM organization. This could be done by comparing the observed phenotypes upon acute CENP-E depletion with unrelated treatments that induce a mitotic delay in the presence of microtubules and active CENP-E (e.g. MG132, Taxol, etc). As currently stated, it sounds like these downstream mitotic defects, present also in patients material with CENP-E mutations, are a consequence of deficient PCM organization. This must be clarified.

4- The localization of CENP-E at the centrosome/PCM/centriolar satellites/microtubules in G2 and Prophase must be determined relative to bona fide markers for each structure at the highest possible resolution. The authors refer to "cytoplasmic" localization in prophase and "enriched around the centrosome" in G2, but it is unclear how these localizations account for the proposed function.

Minor issues:

1- Determination of CENP-E levels by western blot should be performed relative to a serial dilution of control cell extracts. This will allow a more accurate estimation of the depletion efficiency.

2- Introduction: the authors refer that CENP-E contributes to the spindle assembly checkpoint.

However, cells depleted of CENP-E arrest in mitosis, clearly demonstrating a perfectly functional checkpoint. These original ideas were heavily influenced by in vitro work in xenopus extracts and later shown not to be the case in human cells. Whether CENP-E plays a role in MT-kinetochore interactions is also debatable.

3- Results, page 6: "CENP-E KO cells might exit from the cell cycle". Another possibility would be that CENP-E KO cells die in mitosis upon arrest, leading to slow increase in cell number (fig 2d). This should be tested or otherwise the text re-written to reflect this possibility.

4- several occasions throughout the text, the authors refer to metaphase delay after CENP-E depletion. This is incorrect as, by definition, metaphase reflects complete chromosome alignment at the spindle equator, clearly not the case of CENP-E depleted cells. The authors should refer to these as "prometaphase" cells.

5- The proposed link with microcephaly is weak. In fact, only the non-PCM-related phenotypes are observed in patient material and thus are irrelevant for the uncovered CENP-E function in interphase. Either this link is further sustained or I would recommend removing these data.

6- Panels in figures should be increased. It is hard to properly evaluate the quality of the data with such tiny images.

We thank the Reviewer for thoughtful, rigorous reviews. Over the past three months, we have carried out the requested experimentation and have now addressed all comments by Reviewers. **In so doing, we have added 9 new figure panels to the main manuscript and 10 new figure panels to the supplemental figures.** Please see our responses to each Reviewer below in **bold**.

Reviewer #1 (Remarks to the Author):

Major Points:

Fig. 1b: They claim that CENP-E physically interacts with PCM1. It is worth to coimmunostain interphase and mitotic cells with the CENP-E and PCM1 antibodies. **We agree with this statement and have performed the requested experiments. These data, which show co-localization of CENP-E and PCM1, are now presented in new Supplementary Fig. 2.**

Fig. 4e: Immunoblot analysis of PCNT reveals two bands at least; an intact and a cleaved band. An intact band is abundant at G2/M phase while a cleaved band becomes predominant after mitotic exit. The immunoblot data in Fig. 4e are not as expected.

As the Reviewer suggests, Lee et al., 2015 showed abundant levels of cleaved PCNT bands after mitotic exit in HeLa cells. However, we were unable to detect this band in RPE1 cells. As stated in the text, it is likely that the cleaved PCNT is unstable and is degraded quickly after cleavage in RPE1. Thus, there may be cell type-specific differences in stability of cleaved PCNT.

Fig. 5c: The authors conclude that CENP-E physically interacts with PCM1, based on coimmunoprecipitation of endogenous PCM1 with ectopic FLAG-CENP-E. Their conclusion has to be supported by additional experimental evidence, such as definition of the interacting domains and GST pulldown assays.

It is important to point out that we have not over-expressed CENP-E in this experiment: these knock-in cells express native levels of CENP-E as shown in Fig. 2a. Thus, the co-immunoprecipitation has been performed using physiological levels of both proteins. We further respectfully suggest that domain-mapping will not substantially enhance our understanding of this interaction because both proteins are very large with multiple interaction partners, and PCM1 in particular

is a scaffold with many potential interactors (~338 confirmed and experimentally observed interactions for human PCM1 in BioGrid). Thus, the domain mapping will not provide a clear indication of whether the interactions are direct or are mediated by additional proteins and so may not provide substantially new mechanistic insights.

In addition, it is odd that FLAGCENP-E did not coimmunoprecipitate PCM1 in the 12-H cells and monastrol-treated cells.

Based on co-staining of PCM1 and CENP-E (please see new Supplementary Fig. 2), we found that these proteins co-localize within a well-defined window and location: the proteins appeared to be restricted to the centrosome in late S/early G2 phase, but when scattered, PCM1 granules did not fully overlap with CENP-E signal. Therefore, the interactions are indeed highly dynamic, and we have reproducibly shown that PCM1 was not co-precipitated from extracts derived from the 12h time point (most cells are in late G2; please see Supplementary Fig. 1) or mitotic samples.

Fig. 7a: They may examine fragmentation of the spindle poles in the patient cells. **As described in the text, we did not observe PCM fragmentation in mitotic cells from patients, although we did observe a reduction in intensity of PCM at spindle poles. We believe that this variance can be explained by cell type-specific differences or the nature of mutations in the patient cells, although this would need to be tested more extensively in the future.**

We thank the Reviewer for constructive comments that have allowed us to substantially improve our manuscript.

Reviewer #2 (Remarks to the Author):

Major concerns

1. To exclude that the IAA treatment alone may contribute to the observed mitotic phenotypes, control experiments such as doing the same IAA treatment in the cells expressing OsTIR1 with wild-type untagged CENP-E or in the cells with another unrelated degron-tagged protein should be performed (e.g., doing a cell cycle profiling as done in Extended Data Fig. 1).

Importantly, Natsume et al., 2016 clearly showed that TIR1 expression with 0.5 mM IAA treatment did not affect cell growth in human cell lines. More importantly, we also showed that PCM fragmentation was rescued by PCM1 knock-down or expression of constitutively active Plk1. Moreover, we quantitated PCNT S1241 phosphorylation and measured the length of astral MTs after ectopic Plk1(T210D) expression. Importantly, both phenotypes were rescued by expression of constitutively active Plk1, and we now show the results in new Supplementary Fig. 7a-c. Therefore, we conclude that mitotic phenotypes observed in our CENP-E-AID cell lines are specifically due to CENP-E depletion and are not a secondary consequence of IAA treatment.

2. Although compelling genetic evidence has linked centrosome dysfunction to primary autosomal recessive microcephaly, the molecular mechanism underlying the etiology of this disease remains elusive. Studies over the years suggest that premature differentiation and subsequent depletion of neural stem cells (e.g., due to spindle misorientation), apoptosis, cell cycle arrest, neural precursor delamination, or some combinations of these, could be the underlying causes. However, increasing evidence suggests that spindle orientation defects alone cannot explain microcephaly (e.g., Insolera et al., 2014, Jayaraman et al., 2016). Therefore, claiming misorientation of cell division/mitotic spindles upon CENP-E loss as a unifying principle that explains how microcephaly can result from centrosome defects is not well justified.

The Reviewer has raised an important point that we did not consider initially. We now discuss these findings in the text. Jayaraman et al. (2016) showed that *wdr62*^{-/-} mice did not exhibit spindle orientation defects like human cells, suggesting that the mouse brain may have redundant back-up mechanisms for controlling spindle orientation. Indeed, microcephaly in *wdr62*^{-/-} mice appeared milder than expected, and they did not establish *wdr62*^{-/-};*aspm*^{-/-} knock-out mice. Therefore, it is possible that at least one of these proteins is enough to maintain spindle orientation in mice. Insolera et al. (2014) suggested that spindle mis-orientation did not cause microcephaly because knocking out p53 rescued the reduction in brain size of *sas4*^{-/-} mice, but not abnormal spindle orientation. However, it is possible that spindle mis-orientation triggers p53-dependent cell cycle arrest in brain tissues. If this is the case, the rescue of microcephaly with abnormal spindle orientation can be explained. Although we cannot conclude that spindle mis-orientation did not cause microcephaly based on these two papers, we agree that this point requires further

clarification, and we have modified the discussion accordingly (page 16, lines 338-339).

3. The authors observed that in CENP-E knockout cells, PCM1 failed to disperse into the cytoplasm in prophase. They thus conclude that CENP-E transports PCM1 away from the centrosome prior to mitosis. Although this is a very reasonable speculation and CENP-E and PCM1 can be in the same complex (the co-IP experiments), there is no direct evidence to support that CENP-E is indeed the motor that transports PCM1. Other indirect effect of CENP-E on PCM1 localization should also be considered. Or additional data should be obtained to show the direct transport of PCM1 by CENP-E along microtubules.

As the Reviewer states, our co-IP experiments conclusively show that CENP-E and PCM1 interact prior to mitosis, and we have buttressed this finding by adding new data showing co-localization of PCM1 and CENP-E by IF (new Supplementary Fig. 2). Furthermore, and more importantly, we now show that GSK treatment also promotes PCM1 accumulation around centrosomes (new Fig. 6b and d), suggesting that CENP-E motor function is critical for PCM1 redistribution. However, we cannot formally rule out the possibility that other motors, or microtubule reorganization in mitosis, could contribute to the re-distribution in concert with CENP-E, and we have now discussed these alternative explanations in the manuscript (page 13, lines 274-277).

Minor points

1. Fig. 1b, c: How did the authors determine the time point 12h as G2 phase? Is CENP-E really dispersed from the centrosome in prophase? Or does the increase of cytoplasmic CENP-E in prophase mask the CENP-E signals around the centrosome as CENP-E appears to be distributed widely in the cytoplasm, including the pericentrosomal regions?

Based on FACS analysis, the DNA content in the majority of cells reached 4N at 10h after mimosine release, and cells remained 4N after 12h. Using microscopy, we observed few mitotic cells at 12h, and therefore, we conclude that most cells at this time point were in G2 phase. Also, to further demonstrate that the pericentrosomal signal of CENP-E was not masked by the background in prophase, we have included confocal images showing z-stacks around centrosomes (please see new Fig. 1c).

2. Fig. 2: Can the misaligned chromosomes be the result of compromised CENP-E function at BOTH the centrosome and kinetochore? Although the authors emphasize the interphase role of CENP-E, most of the described phenotypes were observed in M phase, where the kinetochore function of CENP-E is expected to be compromised as well.

We thank the Reviewer for this question. Since PCM1 knock-down or Plk1 T210D expression rescued PCM fragmentation, but not chromosome misalignment, in CENP-E KO cells, we conclude that CENP-E regulates distinct events at the centrosome and kinetochore.

3. What happened after cell cycle arrest upon CENP-E loss? Does this arrest eventually trigger apoptosis? Apoptosis of neural progenitors has been linked to the etiology of microcephaly. It thus would be interesting to know if apoptosis follows cell cycle arrest upon CENP-E loss.

To address this interesting and important question, we performed additional FACS analyses after staining cells for AnnexinV to determine whether apoptosis occurs in CENP-E KO cells. We found that most CENP-E KO cells did not undergo apoptosis after cell cycle arrest (please see new Supplementary Fig. 3d). Since patient LCLs do not exhibit cell cycle arrest, we propose that an alternative mechanism (detailed in our model) underlies the disease in these patients, as discussed in the manuscript.

4. Lines 118-121: CENP-E loss could lead to several different mitotic phenotypes, e.g., PCM fragmentation with unfocused spindle poles (Fig. 3a), abnormal astral MTs but with focused spindle poles (Fig. 3c). The time-lapse experiment did not include a MT marker. So it is not clear whether the cells undergoing rotation of metaphase plates are the cells with abnormal astral MTs. This statement should be modified.

We were unable to perform live imaging with multiple channels to visualize MT markers due to low signal-to-noise ratios. We conclude that the rotation was most likely due to the loss of normal astral MTs based on related conclusions regarding phenotypes associated with ASPM knock-down in a previous report (Gai et al., 2016). However, as the reviewer states, we did not provide evidence for a direct correlation between astral MT defects and spindle rotation in our CENP-E KO. Therefore, we have modified the statement accordingly (page 7-8, lines 131-139).

5. The reduction of PLK1 level and S1241 phosphorylated PCNT at the centrosome is very modest. The authors should comment on why the very modest changes could lead to profound mitotic defects such as fragmented PCM.

S1241 phosphorylation of PCNT is critical for PCM expansion. Total loss of the phosphorylation perturbs recruitment of other PCM components to centrosomes, resulting in small bipolar spindles or monopolar spindle formation (Lee and Rhee 2011). On the other hand, almost all of our CENP-E KO cells formed bipolar spindles. In addition, in new Supplementary Fig. 8, we show that astral MT nucleation in the CENP-E KO is normal, implying that the PCM structure is intact until bipolar spindle formation. We propose that the mild PCNT phosphorylation phenotype in CENP-E KO is not sufficient to affect overall PCM expansion but that it impacts the structural stability of PCM after bipolar spindle formation. We have described this possibility in the discussion (page 14, lines 304-307).

6. Fig. 4f: The panels of K82R and T210D conditions should be switched so they will be in the same order as described in the text and in Fig. 4g. In addition, K82R/+dox samples are not the best representative images because the PCNT fragmentation phenotypes are not obvious and seemed to be partially rescued. **We agree with the Reviewer and have made the suggested modifications (please see new Fig. 5f).**

7. Does PLK1(T210D) overexpression also rescue PCNT S1241 phosphorylation, PCNT cleavage defects, astral MT defects, and oblique cell division caused by CENP-E depletion?

To strengthen our conclusions regarding the rescue of phenotypes by Plk1(T210D), we quantitated PCNT S1241 phosphorylation and measured the length of astral MTs after Plk1(T210D) over-expression. Importantly, both phenotypes were rescued by ectopic Plk1(T210D) expression, and we now show the results in new Supplementary Fig. 7a-c.

8. Fig 5. The authors should remind readers that CENP-E-AID fusion contains FLAG tags.

We have made the suggested modification in the text (page 10, line 203).

9. Lines 212 and 228: Compound heterozygous mutations should be a more proper term in describing the D933N/K1355E mutations.

We agree and have made the suggested modification in the text (page 11, line 235).

0. The central piece of the proposed model in this study revolves around the aberrant centrosomal PCM1 accumulation during mitosis upon compromised CENP-E activity, which in turn leads to PCM and cell division defects. The authors should determine if the defects in centrosomal PCNT level and PCNT S1241 phosphorylation in the patient LCLs are indeed caused by the aberrant accumulation of PCM1 at the centrosome. If yes, can PCM1 knockdown also rescue the phenotypes as does in CENP-E KO cells?

To address this question, we performed PCM1 staining in LCLs, and indeed, we observed its centrosomal accumulation in prophase in these patient cells (new Fig. 8g and h). In addition, we performed immunoprecipitations with the CENP-E antibody and found that PCM1 was co-precipitated with CENP-E from wild-type lysates, whereas PCM1 was not detectable in the patient sample (new Supplementary Fig. 9). As suggested, we also attempted to deplete PCM1, but we were unable to knock-down PCM1 expression in LCLs because transfection of these patient cells proved inefficient. Nevertheless, our data from the microscopy and immunoprecipitations experiments suggest that phenotypes in patient LCLs stem from overlapping defects in mechanisms also observed in CENP-E KO cells.

1. Lines 239-246: As mentioned previously, this conclusion is not supported by the data; there is no direct evidence to show that CENP-E transports PCM1 away from the centrosome during G2 phase.

Please see Point #3 above. To address this question, we performed GSK treatment in wild-type cells, and we showed that this treatment also caused PCM1 accumulation in prophase (new Fig. 6b and d). Therefore, CENP-E motor function is critical for PCM1 redistribution. However, it is still possible that other motors (or MT reorganization) act in concert with CENP-E, and we discuss this possibility in the manuscript (page 13, lines 274-277).

2. All characterized CENP-E KO phenotypes are in mitosis. The data did not determine whether these phenotypes are indeed caused by compromised CENP-E function during interphase. Claiming that this kinetochore-independent function of CENP-E is exerted in interphase as an interphase pathway is not well justified.

As described in the manuscript, nearly all CENP-E KO cells form bipolar spindles, implying that PCM fragmentation was driven by tension between the two spindle poles. In other words, centrosomal phenotypes are not obvious until bipolar spindle formation (except abnormal phosphorylation of PCM). Our immunoprecipitation experiments showed that interactions between CENP-E and PCM1 are limited to interphase (Fig. 6e). Further, PCM1 knock-down rescued PCM fragmentation but not chromosome misalignment in CENP-E KO cells. Therefore, we conclude that centrosomal phenotypes in CENP-E KO cells were derived from an interphase pathway.

13. The size of the scale bar in Extended Data Fig. 3b is not indicated.

We apologize for the omission and have now corrected this figure.

We thank this Reviewer for his/her comments which have significantly improved our manuscript.

Reviewer #3 (Remarks to the Author)

Major points

1) related to Figure 3: The authors suggest that PCM fragmentation is a consequence of the inability of CENP-E to remove PCM1 away from the centrosome. The data presented here do not exclude the possibility that PCM fragments due to mitotic arrest. To strengthen the conclusions made upon the data presented in Ext. Data 3a, the authors should have measured the intercentriolar distance. However, the best way to address whether this effect was a consequence of mitotic arrest would be to perform live-cell imaging using fluorescently labeled gamma-tubulin or pericentrin.

We thank the Reviewer for this important suggestion and have therefore performed the following experiments to test the alternative possibility that PCM fragmentation resulted from mitotic arrest. Here, we treated cells with noscapine, which promotes metaphase arrest with misaligned chromosomes, like the CENP-E KO, and examined PCM fragmentation (Fig. 3b and d). Importantly, noscapine treatment induced a slight increase in PCM fragmentation, but the percentage (12%) was significantly lower than in GSK-treated cells (55%). Based on these results, we conclude that PCM fragmentation is not simply a by-product of mitotic delay or arrest.

The effect on astral microtubules should be quantified in a less subjective manner, e.g. by measuring the fluorescence intensity levels. The observation that spindles rotate in the absence of CENP-E (and other proteins important for chromosome congression) is well studied and explained by the effect that the uncongressed chromosomes make on cortical LGN-NuMA-Dynein axis via kinetochore-localized Plk1 (Tame et al., EMBO Rep 2016).

We thank the Reviewer for this suggestion. We quantified the impact of CENP-E ablation on astral MTs by measuring their length and confirmed the dramatic effect on this structure. We have replaced graphs accordingly (please see new Fig. 4b). As the reviewer suggested, Tame et al., EMBO Rep 2016 reported spindle mis-orientation after CENP-E depletion. However, they did not show whether LGN localization was affected by CENP-E depletion. Therefore, it was unclear if spindle mis-orientation in their CENP-E knock-down experiments was fully caused by Plk1 at kinetochores in misaligned chromosomes. We did not observe obvious misaligned chromosomes in patient LCLs, whereas these cells exhibited rotation of metaphase plates at a similar proportion to our CENP-E KO cells, suggesting that malfunction of CENP-E can cause spindle mis-orientation without misaligned chromosomes. Nevertheless, we cite this paper and discuss this possibility in the revised manuscript (page 15, lines 317-323).

2) related to Figure 4: It is important to clarify the cause of reduced levels of centrosomal Plk1. The data presented here do not exclude a possibility that the reduced Plk1 levels are a consequence of problems with centrosome maturation. To properly address this, the fluorescence intensities of gamma tubulin and pericentrin should be quantified in these cells.

We thank the Reviewer for this valuable suggestion. We have now performed the requested quantification and present these data in new Supplementary Fig. 6. CENP-E KO cells did not show reduction in intensities of centrosomal γ -tubulin or PCNT in prophase, suggesting that aberrant phosphorylation of PCNT was due to the reduced centrosomal Plk1 rather than defects in centrosome maturation.

In addition, microtubule nucleation assays should be performed, such as microtubule regrowth assay and quantification of EB1/3 dynamics in interphase and mitosis, which would additionally clarify the impact of CENP-E removal on astral microtubules.

We measured the fluorescent intensity of astral MTs around centrosomes and found that MT nucleation appeared intact in prophase CENP-E KO cells (new Supplementary Fig. 8). Given that almost all CENP-E KO cells formed bipolar spindles, we conclude that the formation of abnormal astral MTs in CENP-E KO cells was not due to impaired MT nucleation ability at spindle poles but rather PCM fragmentation after bipolar spindle formation. We now discuss this conclusion in the revised manuscript (page 10, lines 189-192).

3) related to Figure 5 ??? The authors conclude that CENP-E removes PCM1 away from the centrosomes and that the lack of CENP-E-dependent transport promotes PCM1 accumulation in the vicinity of centrosomes. To dissect whether the observed PCM1 accumulation depended on CENP-E mediated transport, as proposed in the study, or it depended purely on centrosomal localization of CENP-E, an experiment with CENP-E inhibition should be performed. CENP-E inhibition affects the activity of CENP-E, while it should not affect its localization. Thus, if this treatment provides a similar PCM1 accumulation, one could conclude that this is transport-dependent.

We thank the Reviewer for raising a critical point. To address this concern, we have now examined cells after treatment with GSK, which specifically blocks the kinesin motor function of CENP-E. Importantly, we now show that GSK treatment also promotes PCM1 accumulation around centrosomes (please see new Fig. 6b and d), suggesting that CENP-E motor function is critical for PCM1 redistribution. However, we cannot formally rule out the possibility that other motors or microtubule reorganization in mitosis could contribute to the redistribution in concert with CENP-E, and we discuss these alternative explanations in the revised manuscript (page 13, lines 274-277).

The authors show that CENP-E interacts with PCM1 in G2, but not in mitosis. However, all the results are shown in mitotic cells (prophase and metaphase). Do the authors observe any centrosome-related problems in G2?

We did not observe centrosome-related problems in G2 phase. Further, centrosomes in CENP-E KO separated properly in G2 phase, forming two spindle poles in mitosis.

4) related to Figure 6: The authors conclude that the accumulated PCM1 observed

in CENP-E-depleted cells sequesters Plk1 from centrosomes. If true, Plk1 should colocalize with PCM1, which does not seem to be the case.

As the Reviewer points out, we anticipated that Plk1, once sequestered, would colocalize with PCM1, but this was not observed. Our current explanation is that PCM1 is considerably more abundant and is concentrated at satellites, unlike Plk1, which is less abundant and is diffuse. High concentrations of Plk1 would be needed to visualize the protein as foci—as is observed with its centrosomal localization, wherein the protein is restricted to a very small (~100 x 500 nm) region.

As mentioned above, the effect on astral microtubules should be quantified in a less subjective way (fluorescence intensity, EB1/3 dynamics).

We agree with the Reviewer that the impact on astral microtubules should be assessed more objectively. We therefore measured the length of astral MTs before and after CENP-E depletion. The results are shown in new Fig. 4b.

5) related to Figure 7: The link between the patient-derived CENP-E mutant cell line phenotype and the model proposed in this study is poorly established. In contrast to the data obtained by CENP-E degradation, these cells do not show centrosome fragmentation. A slight decrease in pericentrin levels is shown in prophase, but it is not clear whether PCM1 levels and localization is affected, which otherwise constitutes the main phenotype in this study. There is no data provided about this particular CENP-E mutant. The lack of chromosome congression problems suggest that its kinetochore function is intact, but does it localize to centrosome in the same way as the wild type? Does it interact with PCM1? **We showed that PCNT intensity was reduced at spindle poles (current Fig. 8a and b), and PCNT phosphorylation was perturbed in patient LCLs (current Fig. 8c and d). Further, a large proportion of patient LCLs show oblique divisions (new Fig. 8e and f). All of these phenotypes were observed in our CENP-E KO cells. We have also now examined PCM1 staining in these LCLs. Importantly, as observed in our CENP-E KO RPE-1 cells, patient LCLs exhibited PCM1 accumulation around centrosomes in prophase (please see new Fig. 8g and h). In addition, we performed immunoprecipitations with the CENP-E antibody and found that PCM1 was co-precipitated with CENP-E from wild type lysates, whereas PCM1 was not detectable in the patient sample (Supplementary Fig. 9). These data suggest that phenotypes in patient LCLs arise from defects in mechanisms that**

recapitulate those observed in CENP-E KO cells.

Minor points

There are several omissions related to the citation of relevant literature throughout the manuscript. Here are a few examples:

- 1) PCM1 accumulation was previously shown to have a critical impact on centrosome stability, leading to mitotic centrosome fragmentation and unbalanced chromosome segregation (Gr??nb??k Holdgaard et al., Nat Comms 2019).
- 2) As mentioned above, the spindle rotation phenotype in presence of uncongressed chromosomes is explained via its Plk1-dependent effect on cortical LGN-NuMA-Dynein axis (Tame et al. EMBO Rep 2016).
- 3) The possibility of PCM fragmentation by microtubule-induced forces (the discussion section) should be discussed in the context of earlier studies (reviewed in Maiato and Logarinho, NCB 2014).

Based on these suggestions, we have added citations (34, 39-43, 46-49) to the revised manuscript.

We thank this Reviewer for a rigorous review that has significantly improved the manuscript.

Reviewer #4 (Remarks to the Author):

Major issues:

1- In the proposed model, CENP-E microtubule plus-end-directed motor activity would be required to disperse centriolar satellites as cells transit from G2 to mitotic prophase. This conclusion is entirely based on a single experiment (fig 5a,b) that suggests higher accumulation of PCM1 at centrosomes after acute CENP-E-depletion. However, the example shown is clearly not a prophase cell (I would confidently classify it as an early prometaphase cell). I understand that the authors have quantified this in a number of cells, but they must ensure that equivalent and comparable images were used in these quantifications, ideally with proper markers that unequivocally distinguish both states (e.g. including a nuclear envelope marker to confirm that the cell is indeed in prophase).

For these experiments, we selected cells showing chromosome condensation in intact nuclei, and we classified them as prophase cells because we expected that

PCM1 redistribution will have been completed in coordination with this cell cycle stage. Nevertheless, following this reviewer's suggestion we have designated this cell cycle stage as "late prophase/early prometaphase". Also, we have now performed PCM1 staining in GSK-treated cells and observed its accumulation around centrosomes in late prophase/early prometaphase (new Fig. 6b and d), suggesting that CENP-E motor activity is required for PCM1 redistribution.

The proposed model also implies that CENP-E must somehow overcome the activity of the microtubule-minus-end directed motor Dynein, which is required to concentrate centriolar satellites near the centrosome (see Kubo et al., JCB, 1999). Importantly, Dynein and CENP-E activities are differentially regulated by γ -tubulin detyrosination: it inhibits the initiation of Dynein/Dynactin processivity (see e.g. McKenney et al., EMBO J. 2016), while promoting CENP-E motor activity (see Barisic et al., Science, 2015). This means that in order for CENP-E to eventually overcome Dynein, microtubules used in the dispersion of centriolar satellites in prophase must be enriched in detyrosinated tubulin. Is this the case? Please show and discuss.

To address this question, we performed immunofluorescence to detect de-tyrosinated tubulin. We detected this modification exclusively at cilia but did not detect this modification on cytoplasmic or astral MTs in RPE1 cells. This could be explained by cell type-specific or experimental differences.

2- Given that essential microtubule nucleation factors, such as gamma-tubulin, disperse after acute CENP-E depletion, it is expected that microtubule nucleation capacity at centrosomes will be affected (as shown for astral MTs in mitosis). As so, was it the absence of astral microtubules (e.g. due to defective centrosome maturation) that prevented the normal dispersal of centriolar satellites by any CENP-E-independent means, or was it specifically the lack of CENP-E motor activity? This could be assessed by allowing centrosomes to mature in G2 and develop normal astral microtubules and then acutely inhibit CENP-E motor activity with the GSK inhibitor in prophase cells.

We thank the Reviewer for this suggestion. To test whether defects in MT networks affect PCM1 redistribution in CENP-E KO cells, we quantitated the intensity of astral MTs around centrosomes in prophase CENP-E-AID cells and found that MT nucleation in CENP-E KO cells appeared normal. Also, we observed that PCM1 accumulated in GSK-treated cells (new Fig. 6b and d),

suggesting that CENP-E motor function is critical for PCM1 redistribution. However, it is formally possible that other motor proteins or MT reorganization contributes to the redistribution in concert with CENP-E, and we have discussed this possibility accordingly (page 13, lines 274-277).

3-It is arguable that the observed mitotic phenotype results from a direct role of CENP-E in centriolar satellite dispersal, leading to PCM expansion/maturation. Indeed, loss of centrosome integrity and pole splitting, as well as spindle rotation were not rescued by co-depletion of PCM1 and CENP-E, suggesting that these downstream mitotic effects are unrelated to a possible role of CENP-E in PCM organization. Moreover, after acute CENP-E depletion, most cells remain in mitosis for at least 1h, up to 2h. A mitotic delay imposed by unrelated cellular perturbations (including CENP-E inhibition with GSK) is known to generate cohesion fatigue, often followed by loss of centrosome integrity and pole splitting, as well as spindle rotation (see Daum et al., Curr Biol, 2011; Stevens et al., PLoS ONE, 2011). Thus, the authors must exclude that the observed loss of centrosome integrity with consequent pole splitting and spindle rotation is not simply a consequence of the observed mitotic delay imposed by acute CENP-E depletion, rather than through PCM organization. This could be done by comparing the observed phenotypes upon acute CENP-E depletion with unrelated treatments that induce a mitotic delay in the presence of microtubules and active CENP-E (e.g. MG132, Taxol, etc). As currently stated, it sounds like these downstream mitotic defects, present also in patients material with CENP-E mutations, are a consequence of deficient PCM organization. This must be clarified.

We thank the Reviewer for this comment. As suggested by the Reviewer, we performed experiments in which mitotic arrest was induced with noscapine, which promotes metaphase arrest with misaligned chromosomes that mirrors the CENP-E KO cells, and examined PCM fragmentation (new Fig. 3b and d). Noscapine treatment modestly induced PCM fragmentation, but the percentage (13%) was significantly lower than in GSK treated cells (55%). Therefore, we conclude that it is not purely a by-product of mitotic delay or arrest.

4-The localization of CENP-E at the centrosome/PCM/centriolar satellites/microtubules in G2 and Prophase must be determined relative to bona fide markers for each structure at the highest possible resolution. The authors refer to cytoplasmic localization in prophase and enriched around the centrosome

in G2, but it is unclear how these localizations account for the proposed function. **As suggested, we co-stained cells with antibodies against CENPE, CEP164, and γ -tubulin (please see new Fig. 1c). These experiments clearly show the localization of CENP-E at pericentrosomal regions and centriolar satellites. Further, we also performed co-staining of PCM1 and CENP-E and found that these proteins co-localized around centrosomes in late S/early G2 phase, but not in prophase (new Supplementary Fig. 2), suggesting that the interaction is highly dynamic. Taken together with our functional studies--co-immunoprecipitation studies, functional depletion and GSK drug treatments, and comparisons of WT and mutant CENP-E patient cells--we have pinpointed the function of CENP-E to this phase (late S/early G2) of the cell cycle.**

Minor issues:

1-Determination of CENP-E levels by western blot should be performed relative to a serial dilution of control cell extracts. This will allow a more accurate estimation of the depletion efficiency.

We performed the requested experiment, and the data are now shown in new Supplementary Fig. 3b.

2-Introduction: the authors refer that CENP-E contributes to the spindle assembly checkpoint. However, cells depleted of CENP-E arrest in mitosis, clearly demonstrating a perfectly functional checkpoint. These original ideas were heavily influenced by in vitro work in xenopus extracts and later shown not to be the case in human cells. Whether CENP-E plays a role in MT-kinetochore interactions is also debatable.

We agree with these comments and have modified the text accordingly.

3-Results, page 6: CENP-E KO cells might exit from the cell cycle. Another possibility would be that CENP-E KO cells die in mitosis upon arrest, leading to slow increase in cell number (fig 2d). This should be tested or otherwise the text re-written to reflect this possibility.

As shown in Fig. 2c and Fig. 3d, CENP-E KO cells exhibited a mitotic delay, but nearly all of the cells eventually divided, suggesting that they do not die in mitosis. To further test this, we also performed FACS analysis after AnnexinV staining

(please see new Supplementary Fig. 3d) and show that most CENP-E KO cells do not undergo apoptosis.

4- several occasions throughout the text, the authors refer to metaphase delay after CENP-E depletion. This is incorrect as, by definition, metaphase reflects complete chromosome alignment at the spindle equator, clearly not the case of CENP-E depleted cells. The authors should refer to these as prometaphase cells. **We re-phrased the metaphase delay in CENP-E KO as “pseudo-metaphase”, as it was termed in previous studies (Weaver et al., 2003).**

5- The proposed link with microcephaly is weak. In fact, only the non-PCM-related phenotypes are observed in patient material and thus are irrelevant for the uncovered CENP-E function in interphase. Either this link is further sustained or I would recommend removing these data.

In the original version, we showed that PCNT intensity was reduced at spindle poles (current Fig. 8a and b), and PCNT phosphorylation was perturbed in patient LCLs (current Fig. 8c and d). In response to this question, we performed a series of new experiments. First, we performed PCM1 staining and observed that it accumulates around centrosomes in patient LCLs in prophase (Fig. 8g and h), implying that patient LCLs exhibited this phenotype based on defective mechanisms also observed in our CENP-E KO. Second, we performed immunoprecipitations with the CENP-E antibody and found that PCM1 was co-precipitated with CENP-E from wild type lysates, whereas PCM1 was not detectable in the patient sample (Supplementary Fig. 9). Third, patient LCLs show oblique cells division angles, as in the KO. These data suggest that phenotypes in patient LCLs arise from defects in the same mechanisms documented in CENP-E KO cells.

6- Panels in figures should be increased. It is hard to properly evaluate the quality of the data with such tiny images.

We agree and have modified the figures accordingly.

REVIEWERS' COMMENTS:

Reviewer #1 (Remarks to the Author):

The revised manuscript fulfilled all the critics which had been raised by this reviewer.

Reviewer #2 (Remarks to the Author):

The authors did an outstanding job revising this manuscript. I have no further concerns.

Reviewer #3 (Remarks to the Author):

Identification of a non-canonical centrosome-related CENP-E function prior to mitosis is important and I am in general in favor of publishing these data. Although several of my concerns were satisfied in the revised version of manuscript, some important issues remained to be at least discussed or better presented:

- Several figures show tendency to compare prophase with prometaphase cells, instead of prophases alone (e.g. Fig. 6a,b ; Fig. 7a ; Fig. 8c). Since the amount of PCM proteins increase from prophase to prometaphase with centrosome maturation, it is critical not to mix these two phases during quantifications. Therefore, the representative images should present the cells that are without a doubt in the same phase.
- It should be discussed how centrosome maturation is not affected when PCNT is less phosphorylated.
- Noscapine experiment was used to conclude that centrosome fragmentation is more affected by CENP-E inhibition and therefore dependent on CENP-E centrosomal function and not chromosome congression problems. However, the patient cells that do not show chromosome congression/mitotic arrest problems also do not show centrosome fragmentation. This discrepancy should be discussed and the alternative explanations should remained open.
- Since it could not be proved by co-localization that Plk1 is sequestered by PCM1, the conclusion should be toned down more towards "Plk1 being abrogated" or similar.

Reviewer #4 (Remarks to the Author):

The authors are to be congratulated for their effort in addressing the reviewers' concerns. They have now satisfactorily addressed all the points in my original assessment by inclusion of important new data that support the main claims of the work and I now recommend this study for publication.